# A general model to predict small molecule substrates of enzymes based on machine and deep learning

Alexander Kroll[1], Sahasra Ranjan[2], Martin K. M. Engqvist[3,4] & Martin J. Lercher [1] ✉

For most proteins annotated as enzymes, it is unknown which primary and/or secondary reactions they catalyze. Experimental characterizations of potential substrates are time-consuming and costly. Machine learning predictions could provide an efficient alternative, but are hampered by a lack of information regarding enzyme non-substrates, as available training data comprises mainly positive examples. Here, we present ESP, a general machine-learning model for the prediction of enzyme-substrate pairs with an accuracy of over 91% on independent and diverse test data. ESP can be applied successfully across widely different enzymes and a broad range of metabolites included in the training data, outperforming models designed for individual, well-studied enzyme families. ESP represents enzymes through a modified transformer model, and is trained on data augmented with randomly sampled small molecules assigned as non-substrates. By facilitating easy in silico testing of potential substrates, the ESP web server may support both basic and applied science.

Enzymes evolved to efficiently catalyze one or more specific chemical reactions, increasing reaction rates up to over a million-fold over the spontaneous rates[1]. In addition, most enzymes are promiscuous, i.e., they catalyze further, physiologically irrelevant or even harmful reactions[2–4]. Accordingly, a comprehensive mapping of enzyme-substrate relationships plays a crucial role in pharmaceutical research and bio-engineering, e.g., for the production of drugs, chemicals, food, and biofuels[5–7].

Unfortunately, it is both expensive and time-consuming to determine experimentally which reactions are catalyzed by a given enzyme. There is thus a huge imbalance between the number of proteins predicted to be enzymes and the experimental knowledge about their substrate scopes. While the UniProt database[8] contains entries for over 36 million different enzymes, more than 99% of these lack high-quality annotations of the catalyzed reactions. Efforts are

underway to develop high-throughput methods for the experimental determination of enzyme-substrate relationships, but these are still in their infancy[9–11]. Furthermore, even high-throughput methods cannot deal with the vast search space of all possible small molecule substrates, but require the experimenter to choose a small subset for testing.

Our goal in this study was to develop a single machine learning model capable of predicting enzyme-substrate relationships across all proteins, thereby providing a tool that helps to focus experimental efforts on enzyme-small molecule pairs likely to be biologically relevant. Developing such a model faces two major challenges. First, a numerical representation of each enzyme that is maximally informative for the downstream prediction task must be obtained[12]. To be as broadly applicable as possible, these representations should be based solely on the enzymes' primary sequence and not require additional

[1]Institute for Computer Science and Department of Biology, Heinrich Heine University, D-40225 Düsseldorf, Germany. [2]Department of Computer Science and Engineering, Indian Institute of Technology Bombay, Powai, Mumbai 400076, India. [3]Department of Biology and Bioengineering, Chalmers University of Technology, SE-412 96 Gothenburg, Sweden. [4]Present address: EnginZyme AB, Tomtebodavägen 6, 17165 Stockholm, Sweden. ✉e-mail: Martin.Lercher@hhu.de

features, such as binding site characteristics. Second, public enzyme databases only list positive instances, i.e., molecules with which enzymes display measurable activity (substrates)[13]. For training a prediction model, an automated strategy for obtaining suitable negative, non-binding enzyme-small molecule instances must thus be devised.

Existing machine learning approaches for predicting enzyme-substrate pairs were either developed specifically for small enzyme families for which unusually comprehensive training datasets are available[13–18], or they are only capable of connecting substrates with EC classes but not with specific enzymes. For example, Mou et al.[14] developed models to predict the substrates of bacterial nitrilases, using input features based on the 3D-structures and active sites of the enzymes. They trained various machine learning models based on experimental evidence for all possible enzyme-small molecule combinations within the models' prediction scope ($N = 240$). Yang et al.[15] followed a similar approach, predicting the substrate scope of plant glycosyltransferases among a pre-defined set of small molecules. They trained a decision tree-based model with a dataset covering almost all possible combinations of enzymes and relevant small molecules. Pertusi et al.[13] trained four different support vectors machines (SVMs), each for a specific enzyme. As input features, their models only use information about the (potential) substrates, as well as non-substrates manually extracted from the literature; no explicit information about the enzymes was used. Roettig et al.[16] and Chevrette et al.[17] predicted the substrate scopes of small enzyme families, training machine-learning models with structural information relating to the enzymes' active sites. Finally, Visani et al.[19] implemented a general machine-learning model for predicting suitable EC classes for a given substrate. To train this model, all EC classes that are not associated with a certain substrate were used as negative data points, which resulted in a low average positive to negative ratio of 0.0032. Visani et al. did not use any enzyme information beyond the EC class as model input, and therefore the model cannot distinguish between different enzymes assigned to the same EC class.

All these previous models can either not be applied to individual enzymes, or they aim to predict substrates for only a single enzyme or enzyme family. Those models that make predictions for specific enzymes rely on very dense experimental training data, i.e., experimental results for all or almost all potential enzyme-substrate pairs. However, for the vast majority of enzyme families, such extensive training data is not available. As yet, there have been no published attempts to formulate and train a general model that can be applied to predict substrates for specific enzymes across widely different enzyme families. Deep learning models have been used to predict enzyme functions by either predicting their assignment to EC classes[20–22], or by predicting functional domains within the protein sequence[23]. However, different enzymes sharing the same domain architecture or assigned to the same EC class can have highly diverse substrate scopes[24]. Directly predicting specific substrates for enzymes goes an important step beyond those previous methods and can help to predict enzyme function more specifically and more precisely.

Prior work related to the prediction of enzyme-substrate pairs are the prediction of drug-target binding affinities (DTBAs) and of Michaelis-Menten constants, $K_M$ and $k_{cat}$. State-of-the-art approaches in this domain are feature-based, i.e., numerical representations of the protein and the substrate molecule are used as input to machine learning models[25–29]. As numerical descriptions of the substrate molecule, these approaches use SMILES representations[30], expert-crafted fingerprints[31], or fingerprints created with graph neural networks[32,33]. Proteins are usually encoded numerically through deep learning-based representations of the amino acid sequences[34–36]. However, these approaches cannot be transferred one-to-one to the problem of predicting enzyme-substrate pairs. The $K_M$ and $k_{cat}$ prediction models are exclusively trained with positive enzyme-substrate pairs and therefore cannot classify molecules as substrates or non-

substrates[28,29]. Many of the proteins used to train the DTBA prediction models have no enzymatic functions; even if they do, the molecules used for training are mostly not naturally occurring potential substrates, and thus there has been no natural selection for or against binding. In contrast, the binding between enzymes and substrates evolved under natural selection. It appears likely that this evolutionary relationship influences our ability to predict enzyme-substrate pairs, and DTBA models are thus not expected to perform well at this task.

In this work, we go beyond the current state-of-the-art by creating maximally informative protein representations, using a customized, task-specific version of the ESM-1b transformer model[34]. The model contains an extra 1280-dimensional token, which was trained end-to-end to store enzyme-related information salient to the downstream prediction task. This general approach was first introduced for natural language processing[37], but has not yet been applied to protein feature prediction. We created negative training examples using data augmentation, by randomly sampling small molecules similar to the substrates in experimentally confirmed enzyme-substrate pairs. Importantly, we sampled all negative data points from a limited set of metabolites, the set of ~1400 substrates that occur among all experimentally confirmed enzyme-substrate pairs of our dataset. Thus, we do not sample from the space of all possible alternative reactants similar to the true substrates, but only consider small molecules likely to occur in at least some biological cells. While many enzymes are rather promiscuous[2–4], it is likely that most of the potential secondary substrates are not contained in this restricted set for any given enzyme, and hence the chance of sampling false negative data points was likely small. We numerically represented all small molecules with task-specific fingerprints that we created with graph neural networks (GNNs)[38–40]. A gradient-boosted decision tree model was trained on the combined protein and small molecule representations for a high-quality dataset with ~18,000 very diverse, experimentally confirmed positive enzyme-substrate pairs (Fig. 1). The resulting Enzyme Substrate Prediction model – ESP – achieves high prediction accuracy for those ~1400 substrates that have been part of our training set and outperforms previously published enzyme family-specific prediction models.

## Results
### Obtaining training and test data
We created a dataset with experimentally confirmed enzyme-substrate pairs using the GO annotation database for UniProt IDs[41] (Methods, "Creating a database with enzyme-substrate pairs"). For training our machine learning models, we extracted 18,351 enzyme-substrate pairs with experimental evidence for binding, comprised of 12,156 unique enzymes and 1379 unique metabolites. We also extracted 274,030 enzyme-substrate pairs with phylogenetically inferred evidence, i.e., these enzymes are evolutionarily closely related to enzymes associated with the same reactions. These "guilt by association" assignments are much less reliable than direct experimental evidence, and we only used them during pre-training to create task-specific enzyme representations – numerical vectors aimed at capturing information relevant to the prediction task from the enzyme amino acid sequences. Our validations demonstrate that using phylogenetically inferred functions for the construction of appropriate enzyme representations has a positive effect on the prediction of experimentally confirmed enzyme-substrate pairs (see below, "Fine-tuning of a state-of-the-art protein embedding model").

There is no systematic information on negative enzyme-small molecule pairs, i.e., pairs where the molecule is not a substrate of the enzyme. We hypothesized that such negative data points could be created artificially through random sampling, which is a common strategy in classification tasks that lack negative training data[42]. To challenge our model to learn to distinguish similar binding and non-binding reactants, we sampled negative training data only from

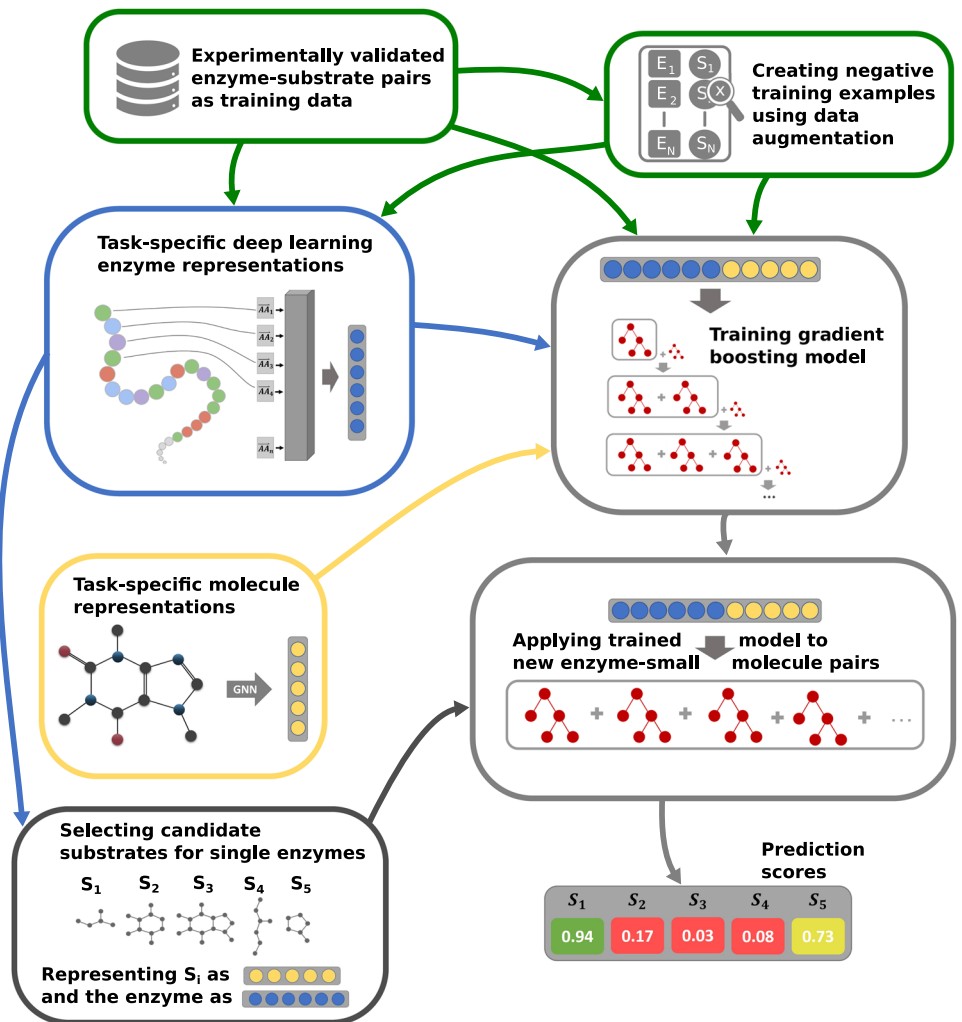

**Fig. 1 | Model overview.** Experimentally validated enzyme-substrate pairs and sampled negative enzyme-small metabolite pairs are numerically represented with task-specific enzyme and small molecule representations. Concatenated enzyme-small molecule representations are used to train a gradient-boosting model. After training, the fitted model can be used to predict promising candidate substrates for enzymes.

enzyme-small molecule pairs where the small molecule is structurally similar to a known true substrate. However, we only considered small molecules included among the experimentally confirmed enzyme-substrate pairs in our dataset. Among such a limited and biased subset, enzymes are quite specific catalysts, and therefore most of the potential secondary substrates are not included for the majority of enzymes. Thus, we assumed that the frequency of incorrectly created negative labels is sufficiently low to not adversely affect model performance. This assumption was confirmed by the high model accuracy on independent test data, as detailed below.

To select putatively non-binding small molecules that are structurally similar to the known substrates, we used a similarity score based on molecular fingerprints, with values ranging from 0 (no similarity) to 1 (identity; see Methods, "Sampling negative data points"). For every positive enzyme-substrate pair, we sampled three molecules with similarity scores between 0.75 and 0.95 to the actual substrate of the enzyme, and used them to construct negative enzyme-molecule pairs. We opted for creating more negative data points than we have positive data points, as this not only provided us with more data, but it also more closely reflects the true distribution of positive and negative data points compared to a balanced distribution.

Our final dataset comprises 69,365 entries. We split this data into a training set (80%) and a test set (20%). In many machine learning

domains, it is standard practice to split the data into training and test set completely at random. However, when dealing with protein sequences, this strategy often leads to test sets with amino acid sequences that are almost identical to those of proteins in the training set. Such close homologs often share the same function[43], and the assessment of model performance could thus be overly optimistic. It is therefore common practice to split such datasets into training, validation, and test sets based on protein sequences similarities[44]. Here, we made sure that no enzyme in the test set has a sequence identity higher than 80% compared to any enzyme in the training set. To show that despite this sequence-based partitioning, enzymes from the training and test sets follow the same distribution, we used dimensionality reduction to map all enzymes to a two-dimensional subspace and plotted the corresponding data points (Supplementary Fig. 1). To evaluate how well our final model performs for different levels of enzyme similarities, we divided the test set further into three subsets with maximal sequence identities between 0–40%, 40–60%, and 60–80% compared to all enzymes in the training set.

**Representing small molecules as numerical vectors**
Extended-connectivity fingerprints (ECFPs) are expert-crafted binary representations for small molecules. The molecules are represented as graphs, with atoms interpreted as nodes and chemical bonds as edges.

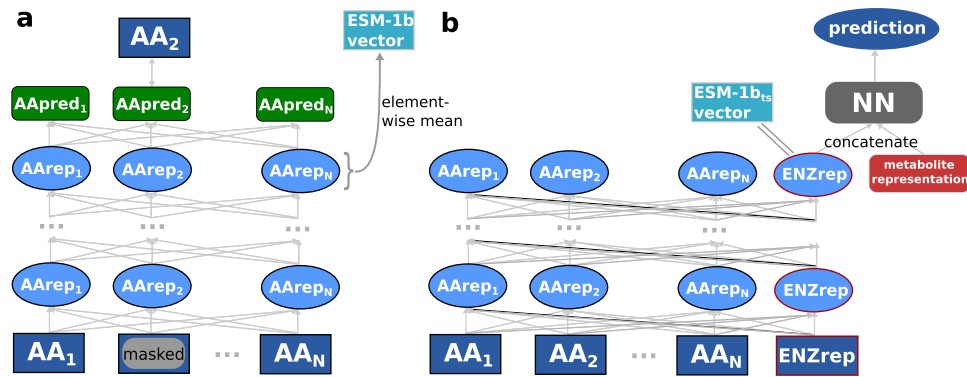

**Fig. 2 | A task-specific enzyme representation developed from the ESM-1b model. a** ESM-1b model. Amino acids of a protein sequence are represented with numerical vectors and passed through a transformer network. Some amino acid representations are masked. All representations are iteratively updated 33 times, using information about neighboring and distant amino acids. The ESM-1b model is trained to predict the masked amino acids. ESM-1b vectors are calculated by taking the element-wise mean of all representations in the last layer. **b** Modified ESM-1b model. An additional representation for the whole enzyme is added to the amino acid representations. After updating all representations 33 times, the enzyme representation is concatenated with a small molecule representation. The network is trained to predict whether the small molecule is a substrate for the given enzyme. After training, the ESM-1b$_{ts}$ vector is extracted as the enzyme representation before adding the small molecule representation.

For the numerical encoding, one classifies bond types and calculates feature vectors with information about every atom (types, masses, valences, atomic numbers, atom charges, and number of attached hydrogen atoms)[31]. Afterwards, these identifiers are updated for a fixed number of steps by iteratively applying predefined functions to summarize aspects of neighboring atoms and bonds. After the iteration process, all identifiers are converted into a single binary vector with structural information about the molecule. The number of iterations and the dimension of the fingerprint can be chosen freely. We set them to the default values of 3 and 1024, respectively. For comparison, we also created 512- and 2048-dimensional ECFPs, but these led to slightly inferior predictions (Supplementary Fig. 2). Using ECFPs can lead to identical representations for structurally very similar molecules, e.g., for some molecules that differ only by the length of a chain of carbon atoms. In our dataset, 182 out of 1379 different molecules shared an identical fingerprint with a structurally similar molecule.

As an alternative to expert-crafted fingerprints such as ECFPs, neural networks can be used to learn how to map graph representations of small molecules to numerical vectors. Such networks are referred to as graph neural networks (GNNs)[38–40]. We trained a GNN for the binary task of predicting if a small molecule is a substrate for a given enzyme. While training for this task, the GNN is challenged to store all information about the small molecule that is relevant for solving the prediction task in a single numerical vector. After training, we extracted these 100-dimensional task-specific vectors for all small molecules in our dataset. It has been observed that pre-training GNNs for a related task can significantly improve model performance[45,46]. Thus, we first pre-trained a GNN for the related task of predicting the Michaelis constants $K_M$ of enzyme-substrate pairs (see Methods, "Calculating task-specific fingerprints for small molecules"). As shown below (see "Successful prediction of enzyme-substrate pairs"), pre-training indeed improved prediction performance significantly. In contrast to ECFPs, GNN-generated fingerprints lead to much fewer cases of identical representations for different molecules. In our dataset, identical fingerprints occurred for 42 out of 1379 molecules.

### Fine-tuning of a state-of-the-art protein embedding model
The ESM-1b model is a state-of-the-art transformer network[47], trained with ~27 million proteins from the UniRef50 dataset[48] in a self-supervised fashion[34]. This model takes an amino acid sequence as its input and puts out a numerical representation of the sequence; these representations are often referred to as protein embeddings. During

training of ESM-1b, ~15% of the amino acids in a protein's sequence are randomly masked and the model is trained to predict the identity of the masked amino acids (Fig. 2a). This training procedure forces the model to store both local and global information about the protein sequence in one 1280-dimensional representation vector for each individual amino acid. In order to create a single fixed-length numerical representation of the whole protein, one typically calculates the element-wise mean across all amino acid representations[34,35,49]. We refer to these protein representations as ESM-1b vectors.

However, simply taking the element-wise mean results in information loss and does not consider the task for which the representations shall be used, which can lead to subpar performance[12]. To overcome these issues, we created task-specific enzyme representations optimized for the prediction of enzyme-substrate pairs. We slightly modified the architecture of the ESM-1b model, adding one additional 1280-dimensional token to represent the complete enzyme, intended to capture information salient to the downstream prediction task (Fig. 2b). The extra token is not adding input information to the model, but it allows an easier extraction of enzyme information from the trained model. This whole-enzyme representation was updated in the same way as the regular ESM-1b amino acid representations.

After a predefined number of update steps, the enzyme representation was concatenated with the small molecule ECFP-vector. The combined vector was used as the input for a fully connected neural network (FCNN), which was then trained end-to-end to predict whether the small molecule is a substrate for the enzyme. This approach facilitates the construction of a single, optimized, task-specific representation. The ESM-1b model contains many parameters and thus requires substantial training data. Therefore, in the pre-training that produces the task-specific enzyme representations, we added phylogenetically inferred evidence to our training set; this resulted in a total of ~287,000 data points used for training the task-specific enzyme representation. After training, we used the network to extract the 1280-dimensional task-specific representations for all enzymes in our dataset. In the following, these representations are called ESM-1b$_{ts}$ vectors.

### Successful prediction of enzyme-substrate pairs
To compare the performances of the different enzyme representations (ESM-1b and ESM-1b$_{ts}$ vectors) and of the two small molecule representations (ECFPs and GNN-generated fingerprints), we estimated prediction quality on our test set when using machine learning models

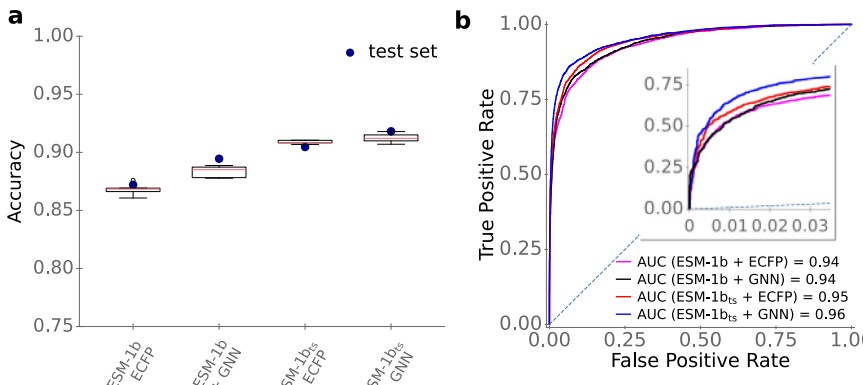

**Fig. 3 | Optimized models provide accurate predictions of enzyme-substrate pairs. a** Accuracies. Boxplots summarize the results of the CV with $n = 5$ folds on the training set with the best sets of hyperparameters. We used a 2 × interquartile range for the whiskers, the boxes extend from the lower to upper quartile values, and the red horizontal lines are displaying the median of the data points. Blue dots display the accuracies on the test set, using the optimized models trained on the whole training set. **b** ROC curves for the test set. The dotted line displays the ROC curve expected for a completely random model. Source data are provided as a Source Data file.

with each of the four combinations of enzyme and small molecule representations. In each case, we concatenated one of the two 1280-dimensional enzyme representations with one of the two small molecule representations to create a single input vector for every enzyme-small molecule pair. We used these inputs to train gradient-boosted decision tree models[50] for the binary classification task of predicting whether the small molecule is a substrate for the enzyme.

We performed hyperparameter optimizations for all four models, including the parameters learning rate, depth of trees, number of iterations, and regularization coefficients. For this, we performed a random grid search with a 5-fold cross-validation (CV) on the training set. To challenge the model to learn to predict the substrate scope of enzymes not included in the training data, we made sure that each enzyme occurred in only one of the five subsets used for cross-validation (Methods, "Hyperparameter optimization of the gradient boosting models"). To account for the higher number of negative compared to positive training data, we also included a weight parameter that lowered the influence of the negative data points. The results of the cross-validations are displayed as boxplots in Fig. 3a. The best sets of hyperparameters are listed in Supplementary Table 1. After hyperparameter optimization, the models were trained with the best set of hyperparameters on the whole training set and were validated on our independent test set, which had not been used for model training or hyperparameter selection. It is noteworthy that for some input combinations, the accuracies on the test set are higher than the accuracies achieved during cross-validation (Fig. 3a). This improved performance on the test set may result from the fact that before validation on the test set, models are trained with approximately 11,000 more samples than before each cross-validation; the number of training samples has a substantial influence on model performance (see below, "Model performance increases with increased training set size").

Commonly used metrics to measure the performance of binary classification models are accuracy, ROC-AUC score, and Matthews correlation coefficient (MCC). Accuracy is simply the fraction of correctly predicted data points among the test data. The ROC-AUC score is a value between 0 and 1 that summarizes how well a classifier is able to distinguish between the positive and negative classes, where a value of 0.5 would result from a model that randomly assigns class labels, and a value of 1 corresponds to perfect predictions. The MCC is a correlation coefficient for binary data, comparable to the Pearson correlation coefficient for continuous data; it takes values between -1 and +1, where 0 would result from a model that randomly assigns class labels, and +1 indicates perfect agreement.

As shown in Fig. 3 and Table 1, models with task-specific enzyme and/or small molecule representations performed better than those with generic representations. The best-performing model combined the fine-tuned ESM-1b$_{ts}$ enzyme representations with the GNN-generated small molecule fingerprints, achieving an accuracy of 91.5%, a ROC-AUC score of 0.956, and an MCC of 0.78. The difference between the two best models (ESM-1b$_{ts}$ + GNN vs. ESM-1b$_{ts}$ + ECFP) is statistically highly significant (McNemar's test: $p < 10^{-5}$). For the final ESP model, we thus chose to represent enzymes with ESM-1b$_{ts}$ vectors and small molecules with GNN-generated, task-specific fingerprints.

To compare the gradient boosting model to alternative machine learning models, we also trained a logistic regression model and a random forest model for the task of predicting enzyme-substrate pairs from the combined ESM-1b$_{ts}$ and GNN vectors. However, these models performed worse compared to the gradient boosting model (Supplementary Table 2).

The GNN used to represent small molecules in the best-performing model was pre-trained for the task of predicting the Michaelis constants $K_M$ of enzyme-substrate pairs. To test if this pre-training improved the predictions, we also tested model performance for fingerprints that were created with a GNN that was not pre-trained. Using a pre-trained GNN indeed led to better model performance (Supplementary Table 3; $p < 10^{-7}$ from McNemar's test).

The results summarized in Table 1 demonstrate that re-training and fine-tuning the ESM-1b model can significantly improve model performance. This finding contrasts previous observations that fine-tuning protein representations can negatively influence model performance and can lead to worse results compared to using the original ESM-1b model[12,51]. To achieve the improved enzyme representations, we added an extra token for the whole enzyme, and we trained the model to store all relevant information for the prediction task in this token. To investigate the importance of the added token for the observed superior performance, we alternatively re-trained the ESM-1b without such an extra token. Our results show that using the extra

**Table 1 | Prediction performance on the test set for all four combinations of enzyme and small molecule representations**

|  | ROC-AUC score | Accuracy | MCC |
|---|---|---|---|
| ESM-1b+ECFP | 0.937 | 87.2% | 0.69 |
| ESM-1b$_{ts}$+ECFP | 0.950 | 90.5% | 0.75 |
| ESM-1b+GNN | 0.940 | 88.8% | 0.72 |
| ESM-1b$_{ts}$+GNN | 0.956 | 91.5% | 0.78 |

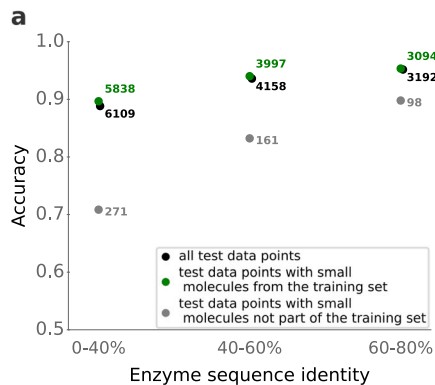

**Fig. 4 | Accurate predictions even for enzymes with distinct sequence similarity compared to enzymes in the training data.** We divided the test set into subsets with different levels of enzyme sequence identity compared to enzymes in the training set. **a** ESP accuracies, calculated separately for enzyme-small molecule pairs where the small molecule occurred in the training set and where it did not occur in the training set. **b** ESP ROC curves. The dotted line displays the ROC curve expected for a completely random model. Source data are provided as a Source Data file.

token indeed improves model performance (Supplementary Table 4; $p = 0.040$ from McNemar's test).

## Good predictions even for unseen enzymes

It appears likely that prediction quality is best for enzymes that are highly similar to enzymes in the training set, and decreases for enzymes that are increasingly dissimilar to the enzymes used for training. How strong is that dependence? To answer this question, we first calculated the maximal enzyme sequence identity compared to the enzymes in the training set for all 2291 enzymes in the test set. Next, we split the test set into three subgroups: data points with enzymes with a maximal sequence identity to training data between 0 and 40%, between 40% and 60%, and between 60% and 80%.

For data points with high sequence identity levels (60-80%), the ESP model is highly accurate, with an accuracy of 95%, ROC-AUC score of 0.99, and MCC of 0.88 (Fig. 4). ESP still performs very well for data points with intermediate sequence identity levels (40–60%), achieving an accuracy of 93%, ROC-AUC score 0.97, and MCC 0.83. Even for enzymes with low sequence identity to training data (0–40%), the ESP model achieves good results and classifies 89% of the data points correctly, with ROC-AUC score 0.93 and MCC 0.72. Thus, while using more similar enzymes during training improves the prediction quality, very good prediction accuracy can still be achieved for enzymes that are only distantly related to those in the training set. The observed differences were statistically significant for sequence identities 0–40% versus 40–60% (Mann-Whitney $U$ test: $p < 10^{-23}$), but not for 40–60% versus 60–80% ($p = 0.14$).

## Low model performance for unseen small molecules

In the previous subsection, we showed that model performance is highest for enzymes that are similar to proteins in the training set. Similarly, it appears likely that the model performs better when making predictions for small molecules that are also in the training set. To test this hypothesis, we divided the test set into data points with small molecules that occurred in the training set ($N = 13,459$) and those with small molecules that did not occur in the training set ($N = 530$).

The ESP model does not perform well for data points with small molecules not present in the training set. When considering only enzyme-small molecules pairs with small molecules not represented in the training set and an enzyme sequence identity level of 0–40% compared to the training data, ESP achieves an accuracy of 71%, ROC-AUC score 0.59, and MCC 0.15. At an enzyme sequence identity level of 40–60%, accuracy improves to 83%, with ROC-AUC score 0.78, and MCC 0.25 for unseen small molecules. At high enzyme sequence identity levels of 60–80%, the accuracy reaches 90%, with ROC-AUC score 0.71, and MCC 0.27. Thus, for unseen small molecules, even a

very moderate model performance requires that proteins similar to the enzyme ( > 40% identity) are present in the training set. We again found the differences to be statistically significant for 0–40% versus 40–60% (Mann-Whitney $U$ test: $p < 10^{-20}$), but not for 40–60% versus 60–80% ($p = 0.226$).

For those test data points with small molecules not present in the training set, we wondered if a high similarity of the small molecule compared to at least one substrate in the training set leads to improved predictions, analogous to what we observed for enzymes with higher sequence identities. For each small molecules not present in the training set, we calculated the maximal pairwise similarity score compared to all substrates in the training set. We could not find any evidence that a higher maximal similarity score leads to better model performance (Supplementary Fig. 3). Hence, we conclude that ESP only achieves high accuracies for new enzyme-small molecule pairs if the small molecule was present among the ~1 400 substrates of our training set.

How many training data points with identical substrates are needed to achieve good model performance? For every small molecule in the test set, we counted how many times the same molecule occurs as an experimentally confirmed substrate in the training set. Supplementary Fig. 4 shows that having as few as two positive training data points for a given small molecule leads to good accuracy when pairing the same small molecule with other enzymes.

## Model performance increases with increased training set size

The previous subsections suggest that a bigger training set with a more diverse set of enzymes and small molecules should lead to improved performance. However, using more data does not guarantee an improved model performance. For example, there could be a limitation in the model architecture that prevents the model from better fitting the data. To test how our model performs with different amounts of training data and to analyze if more data is expected to lead to higher generalizability, we trained the gradient boosting model with different training set sizes, ranging from 30% to 100% of the available training data. Figure 5 shows that accuracy and ROC-AUC score indeed increase with increasing training set size (Spearman rank correlations, accuracy: $\rho^2 = 0.95$, $p < 10^{-4}$; ROC-AUC score: $\rho^2 = 1.0$, $p < 10^{-15}$). Thus, collecting more and more diverse data – for example, through targeted additional experiments – will likely lead to further model improvements.

## ESP can express uncertainty

Internally, our trained classification model does not simply output the positive or negative class as a prediction. Instead, it outputs a prediction score between 0 and 1, which can be interpreted as a

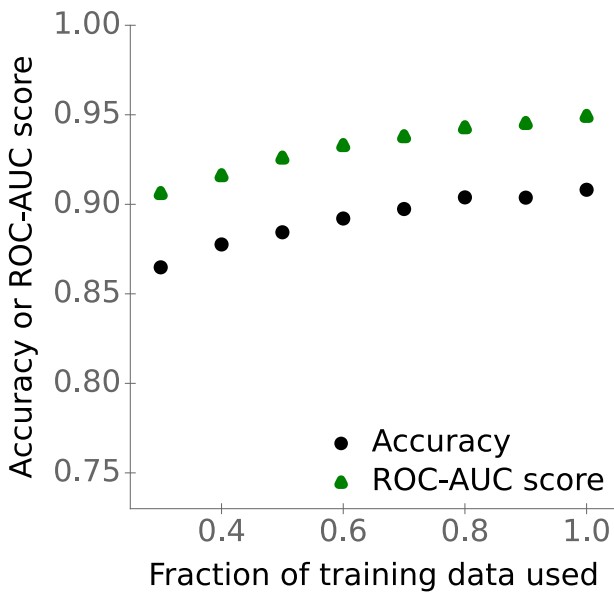

**Fig. 5 | Model performance increases with training set size.** Points show accuracies and ROC-AUC scores for the test set versus the fraction of the available training data used for training the gradient-boosting model. Source data are provided as a Source Data file.

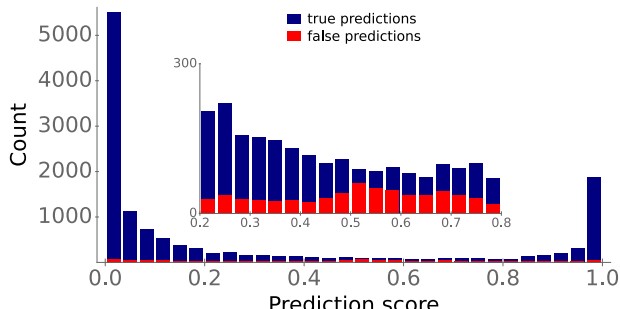

**Fig. 6 | Prediction scores around 0.5 indicate model uncertainty.** Stacked histogram bars display the prediction score distributions of true predictions (blue) and false predictions (red). The inset shows a blow-up of the interval [0.2, 0.8]. Source data are provided as a Source Data file.

measurement of the probability for a data point to belong to the positive class. So far, we assigned all predictions with a score ≥0.5 to the positive class, and all predictions below 0.5 to the negative class. To provide a more detailed view of prediction accuracies, Fig. 6 displays the distributions of the true (blue) and false (red) predictions for our test set across prediction scores.

Most true predictions have a score either close to 0 or close to 1, i.e., the ESP model is very confident about these predictions. In contrast, false predictions are distributed much more evenly across prediction scores. Approximately 4% of prediction scores for our test data fall between 0.4 and 0.6. The model seems to be uncertain for these data points: for this subset, predictions are only barely better than random guesses, with an accuracy of 59%, ROC-AUC score 0.60, and MCC 0.17 (Fig. 6, inset). Thus, when applied in practice, prediction scores between 0.4 and 0.6 should be considered uncertain and should not be assigned to one of the two classes.

## ESP outperforms two recently published models

We compared ESP with two recently published models for predicting the substrate scopes of specific enzyme families. ESP has been trained with much more data points compared to the previously published models; conversely, these previous models used much more detailed input information. Thus, a fair, direct comparison of model architectures is impossible. Instead, we analyzed if our model, which is capable of making use of large amounts of freely available data, can lead to better prediction accuracies than much more targeted approaches that necessarily work on smaller datasets.

Mou et al.[14] trained four different machine learning models (logistic regression, random forest, gradient-boosted decision trees, and support vector machines) to predict substrates of bacterial nitrilases. For model training and validation, they used a dataset with all possible combinations of 12 enzymes and 20 small molecules ($N = 240$), randomly split into 80% training data and 20% test data. We added all training data from Ref.[14] to our training set and validated the updated ESP model on the corresponding test data, which had no overlap with our training data. Mou et al.[14] achieved an accuracy of 82% and a ROC-AUC score of 0.90 on the test set. ESP achieves better results, with an accuracy of 87.5%, ROC-AUC score 0.94, and MCC 0.75.

This improvement is particularly striking given that Mou et al.[14] used knowledge about the enzymes' 3D structures and binding sites, while we only use a representation of the linear amino acid sequences.

Yang et al.[15] published a decision tree-based model, GT-Predict, for predicting the substrate scope of glycosyltransferases of plants. As a training set, they used 2847 data points with 59 different small molecules and 53 different enzymes from *Arabidopsis thaliana*, i.e., the data covered 90.7% of all possible enzyme-small molecule combinations. These authors used two independent test sets to validate the model, one dataset with 266 data points with enzymes from *Avena strigosa* and another dataset with 380 data points with enzymes from *Lycium barbarum*. On those two test sets, GT-Predict achieves accuracies of 79.0% and 78.8%, respectively, and MCCs of 0.338 and 0.319, respectively. We added the training set from Ref.[15] to our training set. The test sets from *Avena strigosa* and *Lycium barbarum* had no overlap with our training data. For these two sets, we achieved similar accuracies as Yang et al. (78.2% in both cases), but substantially improved MCCs: 0.484 for *Avena strigosa* and 0.517 for *Lycium barbarum* (ROC-AUC scores were 0.80 and 0.84, respectively). As the test datasets used by Yang et al.[15] are imbalanced, with a proportion of 18–31% of positive data points, the MCC is a more meaningful score compared to the accuracy[52]; we hence conclude that ESP outperforms GT-Predict. Beyond benchmarking the performance of ESP, the above comparisons of our model predictions to two (almost) complete experimental datasets also indicate that ESP is indeed capable of predicting the full substrate scope of enzymes.

We also tested model performances for the test sets by Mou et al.[14] and Yang et al.[15] without adding any new training data to ESP. Only ~5% and ~8% of the small molecules in these test sets did already occur in our training set. As we showed above that performance drops massively if the model is applied to unseen small molecules (Fig. 4a), we did not expect good model performances. Indeed, for all three test sets, accuracies are below 68%, ROC-AUC scores are below 0.59, and MCCs are below 0.12 (Supplementary Table 5).

## Web server facilitates easy use of ESP

We implemented a web server that allows an easy use of ESP without requiring programming skills or the installation of specialized software. It is available at https://esp.cs.hhu.de. As input, the web server requires an enzyme amino acid sequence and a representation of a small molecule (either as a SMILES string, KEGG Compound ID, or InChI string). Users can either enter a single enzyme-small molecule pair into an online form, or upload a CSV file with multiple such pairs. In addition to the prediction score, the ESP web server reports how often the entered metabolite was present as a true substrate in our training set. Since we have shown that model performance drops substantially when the model is applied to small molecules not used during training, we recommend to use the prediction tool only for

those small molecules represented in our training dataset. We uploaded a full list with all small molecules from the training set to the web server homepage, listing how often each one is present among the positive data points.

## Discussion

Here, we present a general approach for predicting the substrate scope of enzymes; ESP achieves an accuracy of over 91% on an independent test set with enzymes that share at most 80% sequence identity with any enzyme used for training. Notably, the model performs with an accuracy of 89% even for enzymes with very low sequence identity (<40%) to proteins in the training set. This performance seems remarkable, as it is believed that enzymes often evolve different substrate specificities or even different functions if sequence identity falls below 40%[43].

To achieve these results, we use very general input features: a task-specific fingerprint of the small molecule, constructed with a graph neural network (GNN) from a graph representing structural information, and a numerical representation of the enzyme calculated from its amino acid sequence. We show that creating task-specific enzyme representations leads to significant improvements compared to non-task-specific enzyme representations (Fig. 3). One of the major challenges in predicting enzyme-substrate relationships is a lack of experimentally confirmed non-binding enzyme-substrate pairs. To overcome this challenge, we developed a carefully devised strategy of randomly sampling negative enzyme-molecule pairs. Although this data augmentation can potentially lead to false-negative data points, such false negatives are expected to be rare, an expectation that is confirmed by the good results on independent test data sets. Future refinements of this approach might boost model performance further. For example, when creating negative data points for confirmed enzyme-substrate pairs, a tighter decision boundary might result from preferentially choosing structurally similar substrates of highly different enzymes. On the other hand, the sets of true substrates of highly similar enzymes often overlap, and excluding known substrates of highly similar enzymes could avoid creating some false negative data points.

An additional avenue towards potential model improvements could be to test new model architectures. In this study, we trained two separate models for creating task-specific enzyme and small molecule representations. Future work could investigate if the pre-training of the enzyme representation and the small molecule representation could be performed jointly in a single model, thereby creating matched, task-specific enzyme and small molecule representations simultaneously.

Despite the structural similarities of ESP to state-of-the-art models for predicting drug-target binding affinities (DTBAs) and for predicting Michaelis-Menten constants of enzyme-substrate pairs[25–29], the performances of these models are not comparable, as we trained ESP for a binary classification task, whereas the other models address regression tasks. Instead, we compare our approach to two recently published models for predicting enzyme-substrate pairs[14,15]. These two models used very specific input features, such as an enzyme's active site properties and physicochemical properties of the metabolite, and were designed and trained for only a single enzyme family. Our general ESP model – which can be trained on much larger datasets – achieves superior results, despite learning and extracting all relevant information for this task from much less detailed, general input representations. The application of ESP to the dataset from Mou et al.[14] also demonstrate that our model can successfully distinguish between similar potential substrates for the same enzyme, as it achieved good results when it was applied to different nitriles for bacterial nitrilases.

One limitation of ESP is that model performance drops substantially for small molecules that did not occur in the training set.

However, the current version of ESP can still be applied successfully to a broad range of almost 1400 different small molecules present in our dataset. Once more training data becomes available, model performance will very likely improve further (Fig. 5). Mining other biochemical databases–such as BRENDA[53], Sabio-RK[54], and UniProt[8] – for new and non-overlapping data might be a low-cost way to expand the number of different small molecules in the dataset. Adding as few as two additional positive training data points for new molecules will typically lead to accurate predictions (Supplementary Figure 4).

The recent development of AlphaFold[55] and RoseTTAFold[56] facilitates predictions of the 3D structure for any protein with known amino acid sequence. Future work may also include input features extracted from such predicted enzyme structures. Our high-quality dataset with many positive and negative enzyme-small metabolite pairs, which is available on GitHub, might be a promising starting point to explore the utility of such features.

A main use case for the ESP model will be the prediction of possible substrate candidates for single enzymes. In contrast, ESP will likely not lead to satisfactory results when used to predict all enzyme-substrate pairs in a genome-scale metabolic model. This problem results from the trade-off between the True Positive Rate (TPR) and the False Postive Rate (FPR) for different classification thresholds (Fig. 3b). For example, choosing a classification threshold with a TPR of ~80% leads to a FPR of ~5%. If we consider a genome-scale model with approximately 2000 enzymes and 2000 metabolites, then there exist $\sim 4 \times 10^6$ possible enzyme-small molecule pairs, of which only about 6000 will be true enzyme-substrate pairs. A TPR of 80% would lead to the successful detection of 4800 true pairs. At the same time, an FPR of 5% would lead to an additional ~200,000 false predictions.

If, on the other hand, ESP is applied to a set of pre-selected candidate substrates for a single enzyme, a false positive rate of 5% can be acceptable. If we choose 200 molecules as substrate candidates, where one of these 200 is a true substrate for the enzyme, an FPR of 5 % means that the model predicts only ~10 molecules falsely as a substrate, and there is an 80% chance that the true substrate is labeled correctly. This could help to bring down the experimental burden – and associated costs – of biochemical assays to levels where laboratory tests become tractable.

## Methods

### Software
All software was coded in Python[57]. We implemented and trained the neural networks using the deep learning library PyTorch[58]. We fitted the gradient boosting models using the library XGBoost[50].

### Creating a database with enzyme-substrate pairs
To create a database with positive enzyme-substrate pairs, we searched the Gene Ontology (GO) annotation database for UniProt IDs[41] for experimentally confirmed annotations of the catalytic activity of enzymes. A GO annotation consists of a GO Term that is assigned to a UniProt ID, which is an identifier for proteins. GO Terms can contain information about the biological processes, molecular functions, and cellular components in which proteins are involved[59]. We first created a list with all 6587 catalytic GO Terms containing information about enzyme-catalyzed reactions. For each of these GO Terms, we extracted identifiers for the substrates involved in the reaction. If the GO Term definition stated that the reaction is reversible, we treated all reactants (including products) as substrates; if a reaction was labeled as irreversible, we only extracted the reactants annotated as substrates. For this purpose, we used a RHEA reaction ID[60] from the GO Term, which was available for 4086 out of 6587 GO Terms. If no RHEA reaction ID was listed for the GO Term, we extracted the substrate names via text mining from the GO Term definition. Substrate names were then mapped to KEGG and ChEBI identifiers via the synonym database from KEGG[61], or, if no entry in KEGG was found, the PubChem synonym

database[62]. We discarded all 824 catalytic GO Terms for which we could not map at least one substrate to an identifier.

Entries in the GO annotation database have different levels of evidence: experimental, phylogenetically-inferred, computational analysis, author statement, curator statement, and electronic evidence. For training our final model, we were interested only in entries with catalytic GO Terms based on experimental evidence. From these, we removed 6219 enzyme-substrate pairs with water, oxygen, and ions, as these small substrates did not lead to unique representations (see below). We extracted protein and substrate IDs for the remaining 18,351 enzyme-substrate pairs with experimental evidence. 15051 of these pairs resulted from a GO Term that was associated with a RHEA reaction ID, the rest were created via text mining of GO Term definitions. These data points are combinations of 12,156 unique enzymes and 1379 unique substrates.

Before training our models for predicting enzyme-substrate pairs, we pre-trained the ESM-1b protein representations to capture information relevant to enzyme-substrate binding. Due to the high dimensionality of the protein representations, much more data than the 18,351 enzyme-substrate pairs with experimental evidence was required for this task. Only for this pre-training, we thus additionally extracted protein and substrate IDs for 274,030 entries with catalytic GO Terms and phylogenetically inferred evidence (this set excludes 98 384 entries with water, oxygen, and ions as substrates). 200,634 of these enzyme-substrate pairs resulted from a GO Term associated with a RHEA reaction ID, the rest were constructed via text mining of GO Term definitions. These additional data points based on phylogenetic evidence are combinations of 198,259 unique enzymes and 661 unique substrates.

It might be surprising that although we found many more enzyme-substrate pairs with phylogenetically inferred evidence compared to data points with experimental evidence, the number of unique substrates is much smaller. To investigate if we can see a systematic difference between both groups, we plotted the distribution of the first digit of EC classes among the enzymes of both classes. However, no substantial difference was evident except for an over-representation of EC6 (ligases) in the data with phylogenetic evidence (Supplementary Fig. 5). Hence, we assume that the data structure of phylogenetically inferred data points is not an important issue for the calculation of enzyme representations.

We downloaded all enzyme amino acid sequences via the UniProt mapping service[8].

## Sampling negative data points

For every positive enzyme-substrate pair in our dataset, we created three negative data points for the same enzyme by randomly sampling small molecules. The distinction between true and false substrates is harder for small molecules that are similar to the true, known substrates. To challenge our model to learn this distinction, we restricted our sampling of negative data points to small molecules similar to the true substrate. For this purpose, we first calculated the pairwise similarity of all small molecules in our dataset with the function FingerprintSimilarity from the RDKit package DataStructs[63]. This function uses molecular fingerprints of the molecules as its input and computes values between zero (no similarity) and one (high similarity). If possible, we sampled small molecules with a similarity score between 0.7 and 0.95. If we did not find such molecules, we reduced the lower bound in steps of 0.2 until enough small molecules could be sampled. We had to reduce the lower bound in ~19% of enzyme-substrate pairs. We did not simply choose the three most similar compounds as negative data points, because if a substrate appears multiple times in our dataset, this would have led to selecting always the same three small molecules as non-substrates. Instead, we randomly picked three molecules from within the selected similarity range. During this sampling process, we took the distribution of the small molecules among the positive data points into account, i.e., molecules that occur more

frequently as substrates among the positive data points also appear more frequently among the negative data points. To achieve this, we excluded small molecules from the sampling process if these molecules were already sampled enough times (i.e., three times their total occurrence in the set of positive enzyme-substrate pairs).

## Splitting the dataset into training and test sets

Before we split the dataset into training and test sets, we clustered all sequences by amino acid sequence identity using the CD-HIT algorithm[64]. The clusters were created in such a way that two sequences from different clusters do not have a pairwise sequence identity higher than 80%. We used these clusters to split the dataset randomly into 80% training data and 20% test data using a sequence identity cutoff of 80%, i.e., every enzyme in the test set has a maximal sequence identity of 80% compared to any enzyme in the training set. This was achieved by placing all sequences from one cluster either into the training or the test set. To analyze the ESP performance for different sequence identity levels, we further split the test set into subsets with maximal sequence identity to enzymes in the training set of 0–40%, 40–60%, and 60–80% using the CD-HIT algorithm[64].

## Calculating extended-connectivity fingerprints

All small molecules in our final datasets were either assigned to a KEGG ID or ChEBI (Chemical Entities of Biological Interest) ID. For all small molecules with a KEGG ID, we downloaded an MDL Molfile with 2D projections of its atoms and bonds from KEGG[61]. If no MDL Molfile could be obtained in this way, we instead downloaded the International Chemical Idenitifier (InChI) string via the mapping service of MetaCyc[65], if a ChEBI ID was available. We then used the package Chem from RDKit[63] with the MDL Molfiles or InChI strings as the input to calculate the 1024-dimensional binary ECFPs[31] with a radius (number of iterations) of 3. We also calculated 512 and 2048-dimensional ECFPs to investigate if these lead to better model performance than 1024-dimensional ECFPs.

## Calculating task-specific fingerprints for small molecules

In addition to the pre-defined ECFPs, we also used a graph neural network (GNN) to calculate task-specific numerical representations for the small molecules. GNNs are neural networks that can take graphs as their input[38-40]. A molecule can be represented as a graph by interpreting the atoms and bonds of the molecule as nodes and edges, respectively.

We trained and implemented a variant of GNNs called Directed Message Passing Neural Network (D-MPNN)[33], using the Python package PyTorch[58]. To provide the GNN with information about the small molecules, we calculated feature vectors for every bond and every atom in all molecules[28]. For every atom, these features comprise the atomic number, number of bonds, charge, number of hydrogen bonds, mass, aromaticity, hybridization type, and chirality; for every bond, these features comprise bond type, part of ring, stereo configuration, and aromaticity. To input this information into a GNN, the graphs and the feature vectors are encoded with tensors and matrices. While a graph is processed by a GNN, all atom feature vectors are iteratively updated for a pre-defined number of steps by using information of neighboring bond and atom feature vectors. Afterwards, all atom feature vectors are pooled together by applying the element-wise mean to obtain a single graph representation. The dimension $D$ of the updated atom feature vectors and of the final graph representation can be freely chosen; we chose $D = 100$.

This small molecule representation was then concatenated with a small representation of an enzyme; we chose to use a small enzyme representation instead of the full ESM-1b vector to keep the input dimension of the machine learning model used for learning the task-specific small molecule representation low. To compute the small enzyme representation, we performed principal component analysis

(PCA)[66] on the ESM-1b vectors (see below) and selected the first 50 principal components. The concatenated enzyme-small molecule vector was used as the input for a fully connected neural network (FCNN) with two hidden layers of size 100 and 32, which was trained for predicting whether the small molecule is a substrate for the enzyme. We trained the whole model (the GNN including the FCNN) end-to-end. Thereby, the model was challenged to store task-specific and meaningful information in the graph representations. After training, we extracted a graph representation for every small molecule in our training set, which was then used as input for the complete enzyme-substrate pair prediction model.

We performed a pre-training of the described GNN by training it for the related task of predicting the Michaelis constants $K_M$ of enzyme-substrate pairs. As for the task of identifying potential enzyme-substrate pairs, the prediction of $K_M$ is dependent on the interaction between enzymes and small molecules, and hence, this pre-training task challenged the GNN to learn interactions between an enzyme and a substrate. To train the model for the $K_M$ prediction, we used a dataset that was previously constructed for a $K_M$ prediction model[28]. After pre-training, we fine-tuned the GNN by training it for the task of predicting enzyme-substrate pairs, i.e., we used all parameters that were learned during the pre-training task as initial parameters for the GNN that was fine-tuned.

## Calculating enzyme representations

We used the ESM-1b model[34] to calculate 1280-dimensional numerical representations of the enzymes. The ESM-1b model is a transformer network[47] that takes amino acid sequences as its input and produces numerical representations of the sequences. First, every amino acid in a sequence is converted into a 1280-dimensional representation, which encodes the type of the amino acid and its position in the sequence. Afterwards, every representation is updated iteratively for 33 update steps by using information about the representation itself as well as about all other representations of the sequence using the attention mechanism[67]. The attention mechanism allows the model to selectively focus only on relevant amino acid representations to make updates[67]. During training, ~15% of the amino acids in a sequence are masked at random, and the model is trained to predict the type of the masked amino acids. The ESM-1b model has been trained with ~27 million proteins from the UniRef50 dataset[48]. To create a single representation for the whole enzyme, ESM-1b calculates the element-wise mean of all updated amino acids representations in a sequence[34]. We created these representations for all enzymes in our dataset using the code and the trained ESM-1b model provided by the Facebook AI Research team on GitHub.

## Task-specific fine-tuning of the ESM-1b model

To create task-specific enzyme representations for our task of predicting enzyme-substrate pairs, we modified the ESM-1b model. For every input sequence, in addition to the representations of all individual amino acids, we added a token that represents the whole enzyme. This enzyme representation is updated in the same way as the amino acid representations. The parameters of this modified model are initialized with the parameters of the trained ESM-1b model, setting the additional enzyme token initially to the element-wise mean of the amino acid representations. After the last update layer of the model, i.e., after 33 update steps, we take the 1280-dimensional representation of the whole enzyme and concatenate it with a representation for a metabolite, the 1024-dimensional ECFP vector (see above).

This concatenated vector is then used as the input for a fully-connected neural network (FCNN) with two hidden layers of size 256 and 32. The whole model was trained end-to-end for the binary classification task of predicting whether the added metabolite is a substrate for the given enzyme. This training procedure challenged the model to store all necessary enzyme information for the prediction

task in the enzyme representation. After training the modified model, we extracted the updated and task-specific representations, the ESM-1b$_{ts}$ vectors, for all enzymes in our dataset.

We implemented and trained this model using the Python package PyTorch[58]. We trained the model with the extended dataset of 287,386 enzyme-substrate pairs with phylogenetically inferred or experimental evidence for 2 epochs on 6 NVIDA DGX A100s, each with 40GB RAM. Training the model for more epochs did not lead to improved results. Because of the immense computational power and long training times, it was not possible to perform a systematic hyperparameter optimization. We chose hyperparameters after trying a few selected hyperparameter settings with values similar to the ones that were used for training the original ESM-1b model.

## Fine-tuning the ESM-1b model without an additional token

To investigate the effect on model performance of adding a token for the whole enzyme to the ESM-1b model, we also re-trained the model without such an extra token. Instead, we calculated the element-wise mean of all amino acid representations after the last update layer of the model, as is done in the original ESM-1b model. We concatenated the resulting 1280-dimensional vector with a representation for a metabolite, the 1024-dimensional ECFP vector. As for the model described above, this concatenated vector is then used as the input for a fully-connected neural network (FCNN) with two hidden layers of size 256 and 32. The whole model was trained end-to-end for the binary classification task of predicting whether the added metabolite is a substrate for the given enzyme. The training procedure of this model was identical to the model with an additional token for the whole enzyme (see above).

## Hyperparameter optimization of the gradient-boosting models

To find the best hyperparameters for the gradient boosting models, we performed 5-fold cross-validations (CVs). To ensure a high diversity between all folds, we created the five folds in such a way that the same enzyme would not occur in two different folds. We used the Python package hyperopt[68] to perform a random grid search for the following hyperparameters: learning rate, maximum tree depth, lambda and alpha coefficients for regularization, maximum delta step, minimum child weight, number of training epochs, and weight for negative data points. The last hyperparameter was added because our dataset is imbalanced; this parameter allows the model to assign a lower weight to the negative data points during training. To ensure that our model is indeed not assigning too many samples to the over-represented negative class, we used a custom loss function that contains the False Negative Rate, FNR, and the False Positive Rate, FPR. Our loss function, $2 \times FNR^2 + FPR^{1.3}$, penalizes data points that are mistakenly assigned to the negative class stronger than data points that are mistakenly assigned to the positive class. After hyperparameter optimization, we chose the set of hyperparameters with the lowest mean loss during CV. We used the python package xgboost[50] for training the gradient boosting models.

## Displaying the results of cross-validations with boxplots

We used boxplots to display the results of the 5-fold cross-validations, which we performed to find the best set of hyperparameters. We used a 2×interquartile range for the whiskers, the boxes extend from the lower to upper quartile values, and the red horizontal lines are displaying the median of the data points.

## Training of additional machine learning models

To compare the performance of the gradient boosting model to additional machine learning models, we also trained a logistic regression model and a random forest model for the same prediction task. To find the best hyperparameters for the models, we again performed 5-fold CVs on the training set. For the random forest model, the

hyperparameter optimized was the number of estimators, and for the logistic regression model we searched for the best penalty function and coefficient of regularization strength. We used the python package scikit-learn[69] for training both models.

### Validating our model on two additional test sets

We compared the performance of ESP with two published models for predicting the substrate scope of single enzyme families. One of these models is a machine learning model developed by Mou et al. to predict the substrates of 12 different bacterial nitrilases[14]. Their dataset consists of 240 data points, where each of the 12 nitrilases was tested with the same 20 small molecules. This dataset was randomly split by Mou et al. into 80% training data and 20% test data[14]. We added all training data to our training set. After re-training, we validated our model performance on the test set from Ref. 14.

The second model that we compared to ESP is a decision tree-based model, called GT-predict, for predicting the substrate scope of glycosyltransferases of plants[15]. As a training set, Yang et al.[15] used 2847 data points with 59 different small molecules and 53 different enzymes from *Arabidopsis thaliana*. They used two independent test sets to validate model performance: one dataset with 266 data points comprising 7 enzymes from *Avena strigose* and 38 different small molecules, and a second dataset with 380 data points comprising 10 enzymes from *Lycium barbarum* and 38 different small molecules. We added all training data to our training set. After re-training, we validated ESP model performance on both test sets from Ref. 15.

### Analyzing the effect of training set size

To analyze the effect of different training set sizes, we created eight different subsets of our training set, with sizes ranging from 30% to 100% of the original training set size. To create these subsets, we first generated an enzyme list containing all enzymes of the training set in random order. To create the subsets, we extracted all training data points with enzymes that occur in the first 30%, 40%, …, 100% of the generated enzyme list. Afterwards, we re-trained our model on all different subsets of the training set and validated each version on our full test set.

### Statistical tests for model comparison

We tested if the difference in model performance between the two models with ESM-1b$_{ts}$ and ECFP vectors compared to the model with ESM-1b$_{ts}$ vectors and GNN-generated fingerprints is statistically significant. For this purpose, we used McNemar's test[70] (implemented in the Python package Statsmodels[71]), testing the null hypothesis that both models have a similar proportion of errors on our test set. We could reject the null hypothesis ($p < 10^{-9}$), concluding that combining ESM-1b$_{ts}$ vectors with GNN-generated fingerprints leads to a statistically significant improvement over a combination with ECFP vectors. We performed the same test to show that a model with fingerprints created with a pre-trained GNN achieves improved results compared to a model with fingerprints created with a not pre-trained GNN ($p < 10^{-7}$). Moreover, we used McNemar's test to show that the model with ESM-1b$_{ts}$ vectors and GNN-generated fingerprints achieves significantly improved performance compared to the model with ESM-1b and ECFP vectors as the input ($p < 10^{-37}$) and also compared to the model with ESM-1b and GNN-generated fingerprints ($p < 10^{-19}$). Furthermore, we used the same test to show that the task-specific enzyme representations, the ESM-1b$_{ts}$ vectors, that were created by fine-tuning the ESM-1b model with an extra token for the whole enzyme achieved improved performance compared to task-specific enzyme representations that resulted from fine-tuning the ESM-1b model without such an extra token ($p = 0.040$).

We also tested if the differences in model performance between the three different splits of our test set with different enzyme sequence identity levels (0–40%, 40–60%, and 60–80%) are statistically significant. Here, we used the non-parametric two-sided Mann–Whitney $U$ test implemented in the Python package SciPy[72] to test the null hypothesis that the prediction errors for the different splits are equally distributed.

### Reporting summary

Further information on research design is available in the Nature Portfolio Reporting Summary linked to this article.

## Data availability

All data generated in this study and all processed data used to produce the results of this study have been deposited in the GitHub repository available at https://github.com/AlexanderKroll/ESP[73]. Source data for all figures are provided with this paper. Source data are provided with this paper.

## Code availability

The Python code used to generate all results is publicly available only at https://github.com/AlexanderKroll/ESP[73].

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

## Acknowledgements
We thank Veronica Maurino for insightful discussions. Computational support and infrastructure was provided by the "Centre for Information and Media Technology" (ZIM) at the University of Düsseldorf (Germany). We acknowledge financial support to M.J.L. by the German Research Foundation (DFG) through CRC 1310, and, under Germany's Excellence Strategy, through EXC2048/1 (Project ID:390686111), as well as through a grant by the Volkswagen Foundation under the "Life" initiative.

## Author contributions
S.R. implemented and trained the task-specific ESM-1b$_{ts}$ model. A.K. designed the dataset and models and performed all other analyses. M.K.M.E. conceived of the study. M.J.L. supervised the study and acquired funding. A.K., M.K.M.E., and M.J.L. interpreted the results and wrote the manuscript.

## Funding

## Competing interests
The authors declare no competing interests.
