## [Peer Review File · Nature Communications]

REVIEWER COMMENTS

Reviewer #1 (Remarks to the Author):

The problem described in the paper is well-motivated. However, the paper lacks novelty. Importantly, the paper exhibits several serious flaws, in methods and conclusions, that are problematic. Specific concerns:

1. A major concern is the erroneous assumption made when selecting the training. The biological assumption is flawed. Alas, there is sufficient documentation across the field of protein and metabolic engineering that enzymes are not specific: they act on substrates that are similar to ones that they are associated with. The paper by Pertusi et al (Figure 1) that the authors already cite clearly describes the issue of distinguishing between positive and negative data. A recent paper by Visani (<https://doi.org/10.1093/bioinformatics/btab054>) further expands on the issue with training on negative data. There are volumes of machine learning papers to fundamentally discuss this issue, including from the Visani paper: “In machine-learning applications, including such ‘hard negative’ examples can fine-tune the decision boundary between positives and negatives (e.g. Radenovic et al., 2016)”, and how to train when you have unlabeled data (instead of negative data):

<https://link.springer.com/article/10.1007/s10994-020-05877-5>

2. Even if the issue above is addressed, a major concern with this paper overall is the lack of innovation in their model. The deep learning model for learning enzyme representation, there was one small addition of an enzyme representation token to the ESM-1b model. The advantage of using GNNs over fingerprints seem limited and discussed as “laborious”. Have the authors tired pre-training the GNNs, which is now becoming main stream practice? Additionally, it is not clear or justified why the gradient boosting model is better suited for a binary classification task than any other ML models. Further, the evaluation does not justify why this model, or parts of it, is superior to any other models.

3. (A) Yet another major concern is the training/evaluation methodology and the ensuing discussion. In machine learning, it is expected that the test data distribution follows the training data distribution. When the test data deviates from the training data, it is expected to get lower performance on the model (and less confidence in the results). As a matter of fact, it has become standard to partition the data in a random fashion, (i.e. similar distributions), and also on less similar distributions to challenge the model. For example, see the Scaffold split in https://deepchem.readthedocs.io/en/latest/api_reference/splitters.html#scaffoldsplitter

In this regard, the authors note “Hence, to achieve predictions with high accuracies for new enzyme-small molecule pairs, data points with the same small molecule should be present in the training set.” This is clearly a misleading way of addressing and framing the data split issue.

(B) Another shocking result is in Figure 3. How can the model possibly perform better on the test set than on the training set? There is no clear justification of this performance, and usually when models perform better on the test set, they would have “memorized” some of the data that they have seen during practice. Please examine your training/validation/test methodology

(C) The paper compared the results on the models developed by Mou et al. and Yang et al. Mou’s model is trained on N=240 data points, limited to 12 enzymes and 20 molecules. The Mou model was not discussed by the authors; but looks like it encompasses logistic regression, random forest, gradient-boosted decision trees, and support vector machines. To compare 2 machine learning model capabilities: they both need to be trained and evaluated on the same dataset. ESP used a much larger data set. The comparison provided does not allow for a fair comparison.

Other concerns:

4. Showing that the model performance increases with training set size (figure 5) holds little value. Of course, more data results in improved performance. Why would anyone train with less data?

5. Drawing conclusions regarding metrics such as “prediction score” are problematic. A better practice in evaluating model performance is to report the performance across different thresholds or precision@k or rank@k in regards to the interaction or averaged from a molecular or enzyme perspective.

6. The use of 80% maximal sequence identity as a cutoff is not justified

Reviewer #2 (Remarks to the Author):

The paper presents a new method for matching substrates to proteins that takes advantage of the latest deep learning representations and models for proteins. The work is clear, well tested, and, in my view, quite publishable with a bit more tuning. My comments below are primarily for discussion and to point out a few places where more detail might be required.

Extrapolation vs. interpolation is a key issue here. It seems your performance is quite good for small molecules related to training set. This still has many uses and I think that the model is useful, as presented, for connecting known molecules to unknown proteins. Perhaps you might look at the introduction and consider illustrating a main use case or two at the start?

The construction of the training dataset and data augmentation show a great deal of careful thought and good execution.

You use a 1280 dimensional extra token for enzyme data and a similarly large vector for small molecules. Why so large? If space allows could you elaborate on what tests led you to these sizes?

Explicit data augmentation was used, but randomly selected pairs are called negative. What future work could lead to better data augmentation? What other ways might we explore? Finding similar substrates that bind very different proteins and other activities that could tighten boundaries and improve resolution? This is motivated by interest, and I am not suggesting additional work here, just fun speculation in response.

Joint representation of protein and small molecules could be future work. Do you think there could be advantages to having protein and small molecule featurization linked?

[& Related: does using your gradient-boosted decision tree (as opposed to a NN prediction layer) make it harder for you to integrate your featurization step with your classification/ prediction step.]

A major limitation is in the small number of small molecules (metabolites). I wonder if in future work we could find drug-protein or other small molecule data sources to increase the chemical diversity you see.

Did you compare ESM to ESM-1b? Perhaps I missed that in the supplement.

The subheading “ESP is able to say “I don’t know” for some data points” should be a bit less informal. Perhaps use more standard ML language to describe your error estimation.

Data and code availability looks good.

The following is not part of the review, but a sidebar.

In reading your work I thought there was a part of a prior work that might spark some ideas when combined with the ideas in this paper. We showed that we could somewhat automatically identify regions on proteins responsible for functional binding in enzyme, co-factor and regulatory/pathway binding events. <https://www.nature.com/articles/s41467-021-23303-9>

It would be interesting to how knowing the binding sites might help with data augmentation, model structure or training set construction here.

Reviewer #3 (Remarks to the Author):

In this manuscript the authors present a new machine learning model to predict enzyme substrates and reactions based on protein sequence. Overall this is a wonderful paper with a creative approach for getting negative training data. I do worry that the text, as written, will be very alarming to many biologists, while many of the concerns are possibly not significant once one deeply understands the details of the method (and many non-computational readers may not reach this deep understanding). I’ll describe some of my concerns below, which are not necessarily meant as hard criticisms. In fact, I’ll respond to some of my own criticisms explaining why they may not be significant. My hope in doing this is that the authors might be able to edit their manuscript to make it less alarming to the broader research community. I also have questions about some details of the method and training set that I hope the authors can clarify in the paper. Again, I find the work itself to be very exciting and promising and of enormous value and interest to the academic community.

Comments/Questions/Suggestions:

1.) I’m confused by a comment made in the introduction: “These approaches are complementary to the prediction of enzyme-substrate pairs, as the substrate scopes of different enzymes within a given EC class or with a specific domain architecture can be highly diverse”. Arguably, DEEP-EC and

DEEP-GO and similar methods are directly competitive to the author's approach as these methods can predict very specific enzyme functions, with single substrate or near single substrate specificity. I think it is perhaps better to say that the author's approach goes a step beyond these other methods by directly predicting specific substrates and products rather than simply associating enzyme sequences with functional descriptors? At least, the sentence is unclear in its explanation of why these methods are complementary?

2.) "This approach was guided by the biochemical intuition that enzymes are specific catalysts; for any given enzyme, the vast majority of randomly selected small molecules will not be substrates." This is a deeply concerning assumption sitting at the very heart of the author's method. Many enzymes are not actually that specific, and they often have side activities on structurally similar molecules. Arguably, part of the reason we do not observe those side activities is because the substrates in question are not present in the cell. All that said, we also know that the "positive" relationships the authors are using to train do in fact represent the primary evolved function of the enzymes in question and thus likely have much greater activity than the randomly selected alternatives. More importantly, the randomly selected alternatives are pulled from a very limited database of other known enzyme substrates and while similar, may still be quite different from the natural substrate (and thus are even more unlikely to be alternative substrates). Thus, this assumption is not as great a concern as it might initially appear. Basically, the sentence itself is fairly objectionable, while the actual assumption the authors are making is not, and this disconnect could hurt the credibility of the method. Perhaps with edits this could be corrected?

3.) The reliance on the GO-annotation database is concerning. Although the paper indicates they have 100 million evidence associated annotations, the vast majority of those annotations are homology based computational propagations, and thus somewhat useless for the author's purposes as the protein families used for this propagation are not at all isofunctional. The authors do go on to say they only use 13K experimentally validated annotations. This should be clarified as soon as possible to ensure readers that the entire GO-annotation database is not being used. In fact, it would be better to directly say: "We trained our method on ~13K Experimentally validated annotations pulled from the GO-annotation database."

4.) "Thus, we only used these additional data during pre-training to create task-specific enzyme representations, but did not use them for final model training." This sentence is not entirely clear and could use more explanation. What are "task-specific enzyme representations"? The text goes on to explain this, but for anyone lacking a detailed understanding of machine learning methods, this text is very hard to interpret.

5.) "For a small, but ultimately unknown, number of enzyme-small molecule pairs this assumption will result in incorrect negative labels. A priori, the frequency of this occurrence was deemed sufficiently low to not adversely affect model performance. This assumption was confirmed a

posteriori by the high model accuracy on independent test data (see below).” Again, I would object to the word “small” here. The authors really don’t know how common enzyme promiscuity is. Modern metabolomics data is revealing a very large number of previously unobserved and uncharacterized chemistry is happening. What the authors are actually assuming is that enzymes are unlikely to interact with other well known and well characterized metabolites associated with other metabolic reactions that the enzymes were not annotated to perform. This is a much less objectionable assumption (but it’s **not** the assumption you’re stating).

6.) Regarding the construction of the training set from GO-annotation DB, there are sentences like this: “We discarded all catalytic GO Terms for which we could not map at least one substrate to an identifier”. Can the authors break down qualitatively how many GO terms had RHEA IDs (most reliable); how many have text-mined relationships; and how many were discarded? Also, how many proteins are associated with the GO terms in each category (both experimentally verified and phylogenetically inferred)?

7.) How did the authors deal with multi-substrate enzymes, like bimolecular reactions ($A+B \Rightarrow C+D$)? Did the authors only associate reactants to enzymes, or were product relationships included as well? It appears the authors kept all associations, but it would be nice if this was explicitly stated? Also, how many product, reactant associated are there in the training set? How many associations were thrown out due to the filtering of ions and other small molecules.

8.) Why did the number of distinct substrates linked to enzymes dramatically decrease when using the phylogenetically inferred annotations (even though you were working with a vastly larger number of enzymes)? What functions disappeared and why?

9.) Just to be clear, when “creating” their own negative enzyme-substrate associations, the authors selected similar compounds from their training set. From the previous section, this means a total of 1379 substrates? So for each of those 1379 compounds, they selected similar compounds from among the same set of 1379? And occasionally they needed to drop similarity thresholds to get enough examples? How often did this have to be done? How many compounds had enough similar compounds without having to decrease the similarity threshold? Did the authors always select the three most similar compound, or just pick randomly from the similarity range? If random, why not use the most similar compounds?

10.) Why did the authors choose the 80% similarity threshold used when splitting the test and training set? They tried other values (I think). Can this selection be justified?

11.) Are collisions likely to occur with the authors Extended-connectivity fingerprints? For example, between extremely similar molecules? If building these fingerprints for the entire KEGG or MetaCyc, do any collisions occur?

12.) The authors say they did not use phylogenetically inferred annotations in training their methods, but it does appear they used this data to build their ESM-1b. Thus it seems the phylogenetically inferred protein families are possibly somehow cooked into ESM-1b? Is it then possible the method does inherit flaws in this database? It is very difficult for a non-expert in this embedding technology to understand whether or not this is a concern.

13.) When stating the accuracy of the method, the authors should be clear this is only applicable to predicting interactions involving molecules in the training set. This is mentioned later in the discussion, but not really in the abstract or introduction.

Reviewer #1:

The problem described in the paper is well-motivated. However, the paper lacks novelty. Importantly, the paper exhibits several serious flaws, in methods and conclusions, that are problematic. Specific concerns:

1. A major concern is the erroneous assumption made when selecting the training. The biological assumption is flawed. Alas, there is sufficient documentation across the field of protein and metabolic engineering that enzymes are not specific: they act on substrates that are similar to ones that they are associated with. The paper by Pertusi et al (Figure 1) that the authors already cite clearly describes the issue of distinguishing between positive and negative data. A recent paper by Visani (<https://doi.org/10.1093/bioinformatics/btab054>) further expands on the issue with training on negative data. There are volumes of machine learning papers to fundamentally discuss this issue, including from the Visani paper: “In machine-learning applications, including such ‘hard negative’ examples can fine-tune the decision boundary between positives and negatives (e.g. Radenovic et al., 2016)”, and how to train when you have unlabeled data (instead of negative data): <https://link.springer.com/article/10.1007/s10994-020-05877-5>

Response: We now realize – thanks to the Reviewer’s comments – that we did not motivate our strategy for sampling negative data points sufficiently, and that this issue requires more explanation. In particular, our biological assumptions were not explained clearly. We are well aware that enzymes are promiscuous, which is one of the motivations for developing our prediction model. We agree with the Reviewer that sampling structurally similar substrates from the set of all possible substrates would likely result in many false negative data points. However, in this study, we only sampled negative data points from a biased and limited dataset: All substrates that occur among our experimentally confirmed positive enzyme-substrate pairs. This subset comprises only ~1,400 substrates, and most of the metabolites in this biased subset are commonly present in biological cells. It appears likely that due to natural selection on specificity, enzymes are typically quite specific among this limited and biased set of substrates. For this reason, the chance of sampling false negative data points with our strategy is likely much smaller than it would be when sampling similar substrates from the set of all possible substrates. The validity of these assumptions is supported by our model’s high accuracy on independent and experimentally confirmed test data. We apologize for having prompted deep concerns regarding our sampling process by not clearly justifying our strategy of creating negative data points. We hope that the reviewer will find the explanations in our revised manuscript more convincing.

We also thank the reviewer for pointing out our omission of the Visani et al. paper and for drawing our attention to the survey paper for learning from positive and

unlabeled data by Bekker and Davis. The problem discussed in the survey paper differs from the problem that we aimed to solve in an important aspect. Bekker and Davis describe methods that try to find reliable negative data points among unlabeled data, by choosing data points that have the highest dissimilarity compared to the positive data points. However, finding reliable negative data points is not an issue in our study: choosing metabolites that are dissimilar to the true substrates of an enzyme will result in true negative data points with very high probability. In contrast, to challenge our model to achieve a sharper decision boundary, our aim was to find negative data points that are not too distinct from the positive data points. As pointed out by the Reviewer, this strategy was likely to result in some false negatives, but this was a necessary trade-off that we had to accept. The high accuracy of our predictions supports this strategy.

Action: In the revised manuscript, we address the reviewers concerns by motivating the sampling process in much more detail (ll. 87-95 and ll. 124-134). Additionally, we now discuss the Visani et al. paper in the introduction and point out the differences in comparison to our approach (ll. 50-55).

2. Even if the issue above is addressed, a major concern with this paper overall is the lack of innovation in their model. **(a)** The deep learning model for learning enzyme representation, there was one small addition of an enzyme representation token to the ESM-1b model. **(b)** The advantage of using GNNs over fingerprints seem limited and discussed as “laborious”. Have the authors tired pre-training the GNNs, which is now becoming main stream practice? **(c)** Additionally, it is not clear or justified why the gradient boosting model is better suited for a binary classification task than any other ML models. Further, the evaluation does not justify why this model, or parts of it, is superior to any other models.

Response (a): We are surprised that the reviewer is concerned with the novelty and innovation of our manuscript, as we successfully address a previously unsolved prediction problem, providing an accurate, general model for predicting enzyme-substrate pairs. Moreover, we show how the ESM-1b model can be expanded to significantly improve model performance. The original ESM-1b model represents the complete protein through the element-wise mean of the individual amino acid representations. The enzyme summary token, introduced in our work for the first time, is a qualitative extension of the ESM-1b model. It significantly boosts the model’s ability to be trained for specific tasks, as we demonstrate in a new analysis.

Our finding that a fine-tuned enzyme representation significantly improves model performance in itself is interesting, as it runs against recent statements about such fine-tuned representations. For example, Detlefsen et al.

(<https://doi.org/10.1038/s41467-022-29443-w>) state that “fine-tuning can be detrimental to performance”, and in another paper Hsu et al.

(<https://doi.org/10.1038/s41587-021-01146-5>) find that a fine-tuned ESM-1b model does not perform well in comparison with an “augmented” one. In our manuscript, we show how fine-tuning an appropriately extended ESM-1b model for a specific task can be executed successfully, and that it is hence worthwhile to explore if the fine-tuning of protein representations can improve model performance.

Action (a): To test if the additional enzyme representation token indeed improves model performance, we re-trained and fine-tuned the ESM-1b model without this additional token; we found that the additional token indeed leads to a statistically significant improvement in prediction quality (ll. 262-270, ll. 610-619, and Supplementary Table 4). We also emphasize that our findings are in contrast to previous studies that investigated the use of fine-tuned enzyme representations (ll. 262-270).

Response (b): We thank the reviewer for suggesting to pre-train a GNN.

Action (b): We pre-trained a GNN for the related task of predicting Michaelis constants (K_M) of enzyme-substrate pairs. We chose to pre-train the model on this task because the successful prediction of K_M depends on encoding features relevant to the interaction between enzymes and small molecules, and the same features are likely relevant for enzyme-substrate predictions. As suggested by the Reviewer, using small molecule representations created with this pre-trained GNN indeed improved model performance. In the final model, we now use pre-trained GNN representations instead of ECFPs to numerically represent small molecules (ll. 170-183, ll. 566-573, and Supplementary Table 3).

Response (c): In preliminary analyses not reported in our manuscript, we had explored the use of alternative machine learning algorithms and found that they performed worse than a gradient boosting model. In retrospect, we have to agree with the Reviewer that it is better to make such comparisons more systematically and to report them in the paper.

Action (c): We additionally trained a logistic regression model and a random forest model, and we compare the results to the results of the gradient boosting model (ll. 253-256, ll. 638-644, and Supplementary Table 2).

3. (A) **(a)** Yet another major concern is the training/evaluation methodology and the ensuing discussion. In machine learning, it is expected that the test data distribution follows the training data distribution. When the test data deviates from the training data, it is expected to get lower performance on the model (and less confidence in the results). As a matter of fact, it has become standard to partition the data in a random fashion, (i.e. similar distributions), and also on less similar distributions to challenge the model. For example, see the Scaffold split in https://deepchem.readthedocs.io/en/latest/api_reference/splitters.html#scaffoldsplitter .

(b) In this regard, the authors note “Hence, to achieve predictions with high

accuracies for new enzyme-small molecule pairs, data points with the same small molecule should be present in the training set.” This is clearly a misleading way of addressing and framing the data split issue.

Response (a): We regret that we previously did not justify and explain our strategy of splitting the data into training and test sets well enough. We agree that it is standard in most ML domains to split training and test data in a random fashion. However, when dealing with protein sequences, the prediction task becomes too easy if many sequences in the test and validation sets are almost identical to sequences in the training set. Conversely, in practical applications, predictions are often required for protein sequences that have no close homologs that can be used for training. For these reasons, when using protein sequences as model inputs, it is common practice to make splits based on sequence similarities, thereby ensuring that the training and test splits do not contain too many almost identical sequences (see for example: <https://doi.org/10.1186/s12859-019-2932-0>, page 2). Splitting our dataset with a sequence identity threshold of 80% still leads to training and test sets that follow the same distribution (see “Action (a)”).

Action (a): To test if the enzymes from the training and from the test set follow the same distributions, we used dimensionality reduction, mapping all enzymes onto a two-dimensional space. We found no evidence for distinct distributions (Supplementary Fig. 1). We also added a more detailed justification for using a sequence similarity threshold to split the dataset into training and test set (ll. 143-155).

Response (b): We show that our model does not perform well on unseen small molecules. This is not an issue of our training and test split, but a limitation of the prediction model. If we had split the dataset in such a way that the test set only contains small molecules that are also in the training set, we could not have detected this important issue. In practical applications, it is important to be aware of the model limitations – readers might otherwise assume that the model also makes reliable predictions for molecules not included in the training data. However, we agree that the sentence quoted by the Reviewer does not clearly state this issue and could be misunderstood.

Action (b): We reformulated the sentence quoted by the reviewer. We now explain the limitation of our model more clearly (ll. 308-310).

(B) Another shocking result is in Figure 3. How can the model possibly perform better on the test set than on the training set? There is no clear justification of this performance, and usually when models perform better on the test set, they would have “memorized” some of the data that they have seen during practice. Please examine your training/validation/test methodology

Response: The model does not perform better on the test set than of the training set; we apologize for not discussing this explicitly in the previous manuscript. In Figure 3a, the boxplots display the results of our 5-fold cross validation (CV). During CV, the model is validated on data that is originally part of our original training data, but which has not been used to train the model during CV. Thus, the boxplots in Figure 3 display results on data that has not been used for model training. It may still seem surprising that model performance on the total test set tends to be better than the results during CV. A likely reason for this observation is that when validating the model on the test set, more training data (~11,000 additional data points) is used than during CV.

Action: We clarified these issues, and now explain why model performance during CV might be lower compared to model performance on the total test set (ll. 232-237).

(C) The paper compared the results on the models developed by Mou et al. and Yang et al. Mou's model is trained on N=240 data points, limited to 12 enzymes and 20 molecules. The Mou model was not discussed by the authors; but looks like it encompasses logistic regression, random forest, gradient-boosted decision trees, and support vector machines. To compare 2 machine learning model capabilities: they both need to be trained and evaluated on the same dataset. ESP used a much larger data set. The comparison provided does not allow for a fair comparison.

Response: We agree that we should clearly state that we are not performing a direct architecture comparison (where it would be indeed unfair to train with different amounts of training data). Instead, we are performing a comparison of different approaches, where our approach allows to make use of much more training data in comparison to previous models.

Action: We added a paragraph at the beginning of the corresponding Results section to explain that we do not perform a direct architecture comparison (ll. 344-348). We also added a description of the models used by Mou et al. (ll. 349-350).

Other concerns:

4. Showing that the model performance increases with training set size (figure 5) holds little value. Of course, more data results in improved performance. Why would anyone train with less data?

Response: We again have to apologize for not motivating our analyses more explicitly. We did not produce Figure 5 because we wanted to train the model with less data, but to estimate how much we expect model performance to improve if more training data becomes available in the future. While more data is unlikely to lead to worse results, it is not clear that it would lead to meaningful improvements; there could be limitations in the model architecture, or all possible information could already be included in the original training set. Figure 5 shows that this is not the

case, and that additional training data will likely lead to further improvements. Indeed, it is common practice to show how model performance changes with different amounts of training data, and to show that more data leads to higher generalizability. For example, a similar analysis has also been performed in the ESM-1b paper by Rives et al. (<https://doi.org/10.1073/pnas.2016239118>, page 3).

Action: We added additional sentences in the corresponding Results section to clearly motivate why we performed this analysis (ll. 318-321).

5. Drawing conclusions regarding metrics such as “prediction score” are problematic. A better practice in evaluating model performance is to report the performance across different thresholds or precision@k or rank@k in regards to the interaction or averaged from a molecular or enzyme perspective.

Response: As far as we understand this comment, the Reviewer would like to see model performance for different thresholds k , where k is the threshold for classifying data points into the positive and negative classes. Such an analysis was already included in the original submission. Figure 3b and Figure 4b show Receiver Operating Characteristic (ROC) curves, where every data point of a ROC curve results from a different k threshold.

Action: None; as far as we understand the Reviewer’s comment, such an analysis is already included in the manuscript.

6. The use of 80% maximal sequence identity as a cutoff is not justified.

Response: We agree that we should have discussed in more detail why we chose a cutoff of 80% maximal sequence identity.

Action: We now added a paragraph to the Results section where we explain our choice for the similarity threshold of 80% (ll. 143-155; also see our “Action” in response to point 3(A)).

Reviewer #2:

The paper presents a new method for matching substrates to proteins that takes advantage of the latest deep learning representations and models for proteins. The work is clear, well tested, and, in my view, quite publishable with a bit more tuning. My comments below are primarily for discussion and to point out a few places where more detail might be required.

Response: We thank the reviewer for this positive assessment.

1. Extrapolation vs. interpolation is a key issue here. It seems your performance is quite good for small molecules related to training set. This still has many uses and I think that the model is useful, as presented, for connecting known molecules to unknown proteins. Perhaps you might look at the introduction and consider illustrating a main use case or two at the start?

Response: We agree with the reviewer that we should clearly state the limitations of our prediction model in the introduction and that illustrating a potential use case helps to understand how our model can be successfully used.

Action: As suggested, we added a paragraph to the introduction, where we discuss the limitations and a potential use case of our model (ll. 102-109).

2. The construction of the training dataset and data augmentation show a great deal of careful thought and good execution.

Response: We thank the reviewer for this positive comment.

Action: None.

3. You use a 1280 dimensional extra token for enzyme data and a similarly large vector for small molecules. Why so large? If space allows could you elaborate on what tests led you to these sizes?

Response: To create enzyme representations, we used the pre-trained ESM-1b model that has 1280-dimensional tokens. Using the pre-trained network does not allow us to change the token dimension and hence, we also had to set the dimension of the extra token to 1280.

In contrast, the dimension of the ECFPs for the small molecules can be chosen freely. 1024 is the default value for ECFPs. We also tested ECFPs with higher

dimension (2048-dimensional) and lower dimension (512-dimensional), but these led to inferior model performance. However, we did not report these tests in the previous manuscript.

Action: We added a Supplementary Figure showing model performance with ECFPs of different dimensions (Supplementary Fig. 2). We added an explanation that lower and higher dimensions for the ECFPs led to slightly inferior model performance (II. 165-166).

4. Explicit data augmentation was used, but randomly selected pairs are called negative. What future work could lead to better data augmentation? What other ways might we explore? Finding similar substrates that bind very different proteins and other activities that could tighten boundaries and improve resolution? This is motivated by interest, and I am not suggesting additional work here, just fun speculation in response.

Response: We agree that improving the data augmentation process has the potential to further improve model performance. We like your suggestion of finding similar small molecules that are substrates for not highly related enzymes and using these as negative data points. Additionally, we speculate that excluding substrates of highly similar enzymes when sampling negative data points could help to avoid the creation of false negative data points.

Action: We added a paragraph to the Discussion section to discuss how improved data augmentation may have the potential to further improve model performance (II. 405-409).

*5. (a) Joint representation of protein and small molecules could be future work. Do you think there could be advantages to having protein and small molecule featurization linked?
(b) [& Related: does using your gradient-boosted decision tree (as opposed to a NN prediction layer) make it harder for you to integrate your featurization step with your classification/ prediction step.]*

Response (a): Training a joint model (e.g., a Transformer Network) that uses a single string with enzyme and small molecule information as its input is in principle possible. However, Transformer Networks contain an enormous number of parameters and to produce satisfactory results, they often require self-supervised pre-training on large datasets – as in the case of the ESM-1b model. Alternatively, one could combine a pre-trained ESM-1b and a pre-trained ChemFormer

(Transformer Network for small molecules) and fine-tune those models simultaneously to update the enzyme and small molecule representations together.

Action (a): We added a paragraph to the Discussion section to talk about alternative model architectures that could potentially improve model performance (ll. 410-414).

Response (b): Currently, it is not more difficult to integrate the featurization step with the classification step. In our current study, we are training two different models to create enzyme representations and small molecule representations. Because we have those two separate models, both representations must be combined into a new, third model for the classification step. For this third model it does not matter if we use a gradient boosting model or a neural network.

Action (b): None

6. A major limitation is in the small number of small molecules (metabolites). I wonder if in future work we could find drug-protein or other small molecule data sources to increase the chemical diversity you see.

Response: We agree that the number of different substrates in our dataset is a major limitation. We hypothesize that using known drug-protein interactions might negatively affect model performance, because many drugs are inhibiting proteins and do not function as substrates. However, searching additional databases for non-overlapping and new data has the potential to increase the applicability of our prediction model. Such potential databases could be BRENDA, Sabio-RK, and UniProt.

Action: We updated the corresponding part in the Discussion section, now discussing the need of searching for additional data to further increase the applicability of our model (ll. 430-432).

7. Did you compare ESM to ESM-1b? Perhaps I missed that in the supplement.

Response: We assume that the reviewer is referring to the ESM-1v model, which is an ensemble of five different Transformer Networks. Re-training and fine-tuning such an ensemble would be much more difficult than re-training a single model such as the ESM-1b model. Therefore, we decided to work with the ESM-1b model and not to explore the use of EMS-1v.

Action: None.

8. *The subheading “ESP is able to say “I don’t know” for some data points” should be a bit less informal. Perhaps use more standard ML language to describe your error estimation.*

Response: We agree that the subheading was too informal.

Action: We changed the subheading to “ESP can express uncertainty for data points with low prediction accuracy“ (l. 327)

9. *Data and code availability looks good.*

Response: We thank the reviewer for this positive comment.

Action: None.

Reviewer #3:

In this manuscript the authors present a new machine learning model to predict enzyme substrates and reactions based on protein sequence. Overall this is a wonderful paper with a creative approach for getting negative training data. I do worry that the text, as written, will be very alarming to many biologists, while many of the concerns are possibly not significant once one deeply understands the details of the method (and many non-computational readers may not reach this deep understanding). I'll describe some of my concerns below, which are not necessarily meant as hard criticisms. In fact, I'll respond to some of my own criticisms explaining why they may not be significant. My hope in doing this is that the authors might be able to edit their manuscript to make it less alarming to the broader research community. I also have questions about some details of the method and training set that I hope the authors can clarify in the paper. Again, I find the work itself to be very exciting and promising and of enormous value and interest to the academic community.

Response: We thank the reviewer for this positive assessment.

Comments/Questions/Suggestions:

1. *I'm confused by a comment made in the introduction: "These approaches are complementary to the prediction of enzyme-substrate pairs, as the substrate scopes of different enzymes within a given EC class or with a specific domain architecture can be highly diverse". Arguably, DEEP-EC and DEEP-GO and similar methods are directly competitive to the author's approach as these methods can predict very specific enzyme functions, with single substrate or near single substrate specificity. I think it is perhaps better to say that the author's approach goes a step beyond these other methods by directly predicting specific substrates and products rather than simply associating enzyme sequences with functional descriptors? At least, the sentence is unclear in its explanation of why these methods are complementary?*

Response: We agree that the sentence on its own was not clear enough and required more explanation.

Action: We now reformulated the paragraph in the introduction to explain more precisely how our method is going a step beyond previous methods (ll. 64-67).

2. *"This approach was guided by the biochemical intuition that enzymes are specific catalysts; for any given enzyme, the vast majority of randomly selected small molecules will not be substrates." This is a deeply concerning assumption sitting at the very heart of the author's method. Many enzymes are not actually*

that specific, and they often have side activities on structurally similar molecules. Arguably, part of the reason we do not observe those side activities is because the substrates in question are not present in the cell. All that said, we also know that the “positive” relationships the authors are using to train do in fact represent the primary evolved function of the enzymes in question and thus likely have much greater activity than the randomly selected alternatives. More importantly, the randomly selected alternatives are pulled from a very limited database of other known enzyme substrates and while similar, may still be quite different from the natural substrate (and thus are even more unlikely to be alternative substrates). Thus, this assumption is not as great a concern as it might initially appear. Basically, the sentence itself is fairly objectionable, while the actual assumption the authors are making is not, and this disconnect could hurt the credibility of the method. Perhaps with edits this could be corrected?

Response: We are grateful to the Reviewer for pointing out that our previous description of sampling negative data points might sound concerning to many readers. We agree that it requires more explanation and justification to show that our sampling approach is unlikely to lead to many false negative data points.

Action: In the revised manuscript, we motivate the sampling process of negative data points in more detail. In the introduction (ll. 87-95) as well as the Results section (ll. 124-134), we reformulated the corresponding paragraphs. We now describe and explain that although many enzymes are quite promiscuous, sampling from our limited dataset of ~1400 substrates, which are commonly present in cells, will likely not lead to many false negative data points.

3. The reliance on the GO-annotation database is concerning. Although the paper indicates they have 100 million evidence associated annotations, the vast majority of those annotations are homology based computational propagations, and thus somewhat useless for the author’s purposes as the protein families used for this propagation are not at all isofunctional. The authors do go on to say they only use 13K experimentally validated annotations. This should be clarified as soon as possible to ensure readers that the entire GO-annotation database is not being used. In fact, it would be better to directly say: “We trained our method on ~13K Experimentally validated annotations pulled from the GO-annotation database.”

Response: We agree that we should directly state that we only used those entries from the GO annotation database with experimental evidence for model training.

Action: We adjusted the first sentence where we stated that we used the data from the GO annotation database to clarify that we only used experimentally validated annotations (ll. 113-115).

4. *“Thus, we only used these additional data during pre-training to create task-specific enzyme representations, but did not use them for final model training.” This sentence is not entirely clear and could use more explanation. What are “task-specific enzyme representations”? The text goes on to explain this, but for anyone lacking a detailed understanding of machine learning methods, this text is very hard to interpret.*

Response: We thank the reviewer for pointing out that the meaning of “task-specific enzyme representations” may be unfamiliar to many readers.

Action: We added an additional sentence that shortly explains what task-specific enzyme representations are, and we added a reference to the corresponding section, where we describe these representations in detail (ll. 118-123).

5. *“For a small, but ultimately unknown, number of enzyme-small molecule pairs this assumption will result in incorrect negative labels. A priori, the frequency of this occurrence was deemed sufficiently low to not adversely affect model performance. This assumption was confirmed a posteriori by the high model accuracy on independent test data (see below).” Again, I would object to the word “small” here. The authors really don’t know how common enzyme promiscuity is. Modern metabolomics data is revealing a very large number of previously unobserved and uncharacterized chemistry is happening. What the authors are actually assuming is that enzymes are unlikely to interact with other well known and well characterized metabolites associated with other metabolic reactions that the enzymes were not annotated to perform. This is a much less objectionable assumption (but it’s *not* the assumption you’re stating).*

Response: We thank the reviewer for pointing out the inaccuracy of our previous phrasing, and that our assumption that we are sampling only a small number of false negative data points was previously not justified enough and required additional explanation.

Action: We added a more detailed description, discussing and explaining why it is likely that only a small number of the sampled small molecules will lead to false negative data points (ll. 124-134).

6. *Regarding the construction of the training set from GO-annotation DB, there are sentences like this: “We discarded all catalytic GO Terms for which we could not map at least one substrate to an identifier”. Can the authors break down qualitatively how many GO terms had RHEA IDs (most reliable); how many have text-mined relationships; and how many were discarded? Also, how many proteins are associated with the GO terms in each category (both experimentally verified and phylogenetically inferred)?*

Response: We agree that we should have broken down the contents of the GO database with respect to annotation quality relative to our task.

Action: As suggested, we added the following numbers in the Methods sections “Creating a database with enzyme-substrate pairs” (ll. 464-491): number of GO Terms containing information about enzyme-catalyzed reactions; number of GO Terms associated with RHEA IDs; number of discarded GO Terms; number of experimentally verified/phylogenetically inferred data points with GO Terms with and without RHEA IDs.

7. *How did the authors deal with multi-substrate enzymes, like bimolecular reactions ($A+B \Rightarrow C+D$)? Did the authors only associate reactants to enzymes, or were product relationships included as well? It appears the authors kept all associations, but it would be nice if this was explicitly stated? Also, how many product, reactant associated are there in the training set? How many associations were thrown out due to the filtering of ions and other small molecules.*

Response: Our aim was to include all reactants that are likely to be substrates of an enzyme at least under some conditions. Thus, if it was stated in the GO Term definition that the enzymatic reaction is reversible, we added all substrates and all products. However, if the reaction was stated to be irreversible, we only added reactants annotated as substrates.

We filtered out 6219 enzyme-substrate combinations, because the substrate was water, oxygen, or an ion.

Action: We now clearly state which enzyme-reactant pairs were extracted from the GO annotation database (ll. 466-468). We also state how many enzyme-substrate combinations were removed, because the substrate was either water, oxygen, or an ion (ll. 477-478).

8. *Why did the number of distinct substrates linked to enzymes dramatically decrease when using the phylogenetically inferred annotations (even though you were working with a vastly larger number of enzymes)? What functions disappeared and why?*

Response: We agree that it is surprising that the number of unique substrates is much smaller among phylogenetically inferred data points than among data points with experimental evidence. We double-checked to confirm that the stated numbers are indeed correct. We have no explanation for this pattern, except that it must be a feature of how people at GO created the pipeline for making phylogenetic inferences.

Action: We added a paragraph to the Methods section discussing the issue of fewer unique substrates among phylogenetically inferred annotations (ll. 492-498). We investigated if the GO Annotation database contains phylogenetically inferred predictions preferably for specific EC classes (Supplementary Fig. 5). However, we could not see any significant differences.

9. *Just to be clear, when “creating” their own negative enzyme-substrate associations, the authors selected similar compounds from their training set. From the previous section, this means a total of 1379 substrates? So for each of those 1379 compounds, they selected similar compounds from among the same set of 1379? And occasionally they needed to drop similarity thresholds to get enough examples? How often did this have to be done? How many compounds had enough similar compounds without having to decrease the similarity threshold? Did the authors always select the three most similar compounds, or just pick randomly from the similarity range? If random, why not use the most similar compounds?*

Response: We did not always pick the three most similar compounds, because for a specific positive substrate, this would always have led to the same negative molecules. Instead, we randomly picked from molecules within a similarity range. We had to reduce the lower bound in ~19% of all cases.

Action: We now describe the sampling process in the Methods section in more detail (ll. 510-514).

10. *Why did the authors choose the 80% similarity threshold used when splitting the test and training set? They tried other values (I think). Can this selection be justified?*

Response: When working with protein sequences, it is standard to split the dataset into training and test set based on a similarity threshold. We chose a similarity threshold of 80% to avoid nearly identical enzymes in the training and the test set. We did not try different values, but we later divided the test set into three subsets (0-40%, 40-60%, 60-80%) to further investigate model performance for different levels of sequence similarities. We thank the reviewer for pointing out that we previously did not justify our choice of similarity threshold well enough.

Action: We now added a paragraph to the Results section where we explain our choice for the similarity threshold of 80% (ll. 143-155).

11. *Are collisions likely to occur with the authors Extended-connectivity fingerprints? For example, between extremely similar molecules? If building these fingerprints for the entire KEGG or MetaCyc, do any collisions occur?*

Response: We indeed found that collisions do occur when using extended-connectivity fingerprints (ECFPs), e.g., between similar molecules with long chains of carbon atoms. Using ECFPs led to collisions for 182 out of 1379 molecules in our dataset.

Action: Following a suggestion of Reviewer 1, we now pretrained a graph neural network (GNN) to create task-specific small molecule representations, and this indeed led to improved results. In the revised manuscript, we thus use GNNs instead of ECFPs to represent small molecules (ll. 170-183 and ll. 566-573). These GNN-generated fingerprints are more specific, and they only lead to identical molecule representations in 42 out of 1379 cases, which we now state explicitly (ll. 172-183).

12. *The authors say they did not use phylogenetically inferred annotations in training their methods, but it does appear they used this data to build their ESM-1b. Thus it seems the phylogenetically inferred protein families are possibly somehow cooked into ESM-1b? Is it then possible the method does inherit flaws in this database? It is very difficult for a non-expert in this embedding technology to understand whether or not this is a concern.*

Response: When using phylogenetically-inferred functions for creating enzyme representations, it could in principle happen that wrongly inferred functions affect the quality of the enzyme representations. However, our evaluation on experimentally validated enzyme-substrate pairs shows that adding phylogenetically inferred data points to the construction of enzyme representations improves model performance. Hence, adding phylogenetically inferred functions in the process of creating enzyme-representations does not appear to be an issue.

Action: We now added an explanation of these issues to the corresponding Results section, emphasizing that using phylogenetically-inferred functions for creating enzyme representations should be unproblematic (ll. 118-123).

13. *When stating the accuracy of the method, the authors should be clear this is only applicable to predicting interactions involving molecules in the training set. This is mentioned later in the discussion, but not really in the abstract or introduction.*

Response: We agree that we should already state in the introduction that model performance is only high if the model is applied to those ~1400 small molecules in our training set.

Action: We now added this information to the Abstract (l. 15) and the Introduction (ll. 100-101).

Overall concern remains:

- The lack of novelty. ESMB-1b was used prior; language models with fingerprints/GNNs were used before; the token is not added information and does not provide significant improvement over not using it. Further, **it actually promotes incorrect ideas about how adding redundant information is helpful in machine learning tasks.**
- Evaluation of the model remains limited.
 - No detailed comparison with other methods. By their own admission the comparison provided is not fair. Why not compare against state-of-the-art methods in predicting interactions (similar to ones that are used for drug-target interactions – most of those are downloadable and readily available)
 - Not using other datasets to show the generalizability of the model
 - The experimental methodology for evaluating the model does not properly evaluate impact of negative data selection nor different data splits, and reporting test data along with cross validation is problematic from a reader's perspective.

Machine learning models are developed to have predictive power. The authors, by their own admission, "**conclude that ESP only achieves high accuracies for new enzyme-small molecule pairs if the small molecule was present among the ~1 400 substrates of our training set.**". The model predictive capabilities are therefore limited. From a machine-learning perspective, this should not be the case, provided that the training data is selected more carefully.

In response to the specific rebuttals:

1. Selection of the "negative data". This concern is not addressed, and the justification provided is not scientifically sound.

The authors do not provide sufficient justification WHY their negative sampling methodology is valid.

They state, "This subset comprises only ~1,400 substrates, and most of the metabolites in this biased subset are commonly present in biological cells." Looking at their Methods section in the paper, the positive interactions were retrieved from enzyme-substrate pairs listed with catalytic activity in UniProt through RHEA. The ABSENCE OF AN INTERACTION DOES NOT MEAN THAT IT IS NEGATIVE, NO MATTER HOW POPULAR THE SUBSTRATE IS. Further, RHEA, per their website, "Rhea is an expert-curated knowledgebase of chemical and transport reactions of biological interest" – So not clear how extensive their curation efforts are towards promiscuous activities.

So, there has to be a way of sampling the negative data and giving them "less confidence". Exploring the ratio of the positives and negatives, while balancing the classes would be a route towards addressing this problem.

2a. The novelty of the ESP model remains a concern based on the author's evaluation.

Thank you for adding the analysis to clarify the impact of this contribution. I am examining your results that you added:

Supplementary Table 4. Results of three gradient boosting models with different enzyme representations. Models were trained with GNN-generated fingerprints as small molecule representations combined with three different enzyme representations: *ESM-1b* vectors, *ESM-1b_{ts}* vectors created without an extra token for the whole enzyme, and *ESM-1b_{ts}* vectors created with an extra token for the whole enzyme. Results are shown for the test set. The hyperparameter optimizations for all models were performed with 5-fold cross-validation on the training set.

	ROC-AUC score	Accuracy	MCC
ESM-1b	0.940	88.8%	0.72
ESM-1b_{ts} (mean representation)	0.956	90.9%	0.77
ESM-1b_{ts} (enzyme token)	0.956	91.5%	0.78

37/42

It looks like your improvement from fine-tuning (a standard practice in machine-learning, not novel) is in the top red box (*ESM-1b_{ts}*) is roughly 2%.

The additional novelty that you are claiming by adding the enzyme token is highlighted in the bottom redbox: that gives you no improvement in ROC-AUC, and less than 1% in Accuracy, and less than 0.01 in MCC, over the fine-tuning method.

The impact of accessing the enzyme token is not significant, per your own results.

From a machine learning perspective, the knowledge in the “enzyme token” should have been already captured by the equivalent representation of the individual amino acids. You are not adding any new information, but summarizing the info for the model ... If the model parameters were further trained and fine-tuned, then it is likely that you would have achieved the additional results without the enzyme token.

3A(a). **Concern about training under different data splits was not addressed.** This reviewer was not asking for data distribution analysis - they are coming from the same distribution. What was being asked is to consider different data splits to show how the model performs on a more challenging split. There is no need for Supplementary Fig. 1. But please examine the different data splits on the molecules.

3A(b) Thank you for rephrasing the sentence. **How it is stated now shows the severe predictive limitations of your models, which is now the main concern regarding this manuscript.**

3(B) **It is not standard practice to have a test set and do cross validation at the same time!** Following standard machine learning reporting practices would avoid this issue. Please remove the unnecessary explanation and follow standard machine learning reporting practices

3(C) Thank you for addressing the comparison with Mou's. However, with you realizing that it is not a fair comparison, **there is currently no fair comparison with any state of the art models on the same dataset.** This becomes yet another concern for the novelty and value of this paper.

4. Unfortunately, the section of training with increasing data set size remains not interesting as the used dataset is *small* (69,365 interactions), and there is no reason to train with less data. In contrast, the ESM-1b Rives paper trains with **250 MILLION** sequences and that warrants exploring training with less data.

5. The "k" comes from retrieval metrics in machine learning. A protein in the testset interactions with k molecules. You want to rank the model's ability in ranking these k interactions ahead of all other interactions in the dataset. See wikipedia definitions:
[https://en.wikipedia.org/wiki/Evaluation_measures_\(information_retrieval\)#Precision_at_k](https://en.wikipedia.org/wiki/Evaluation_measures_(information_retrieval)#Precision_at_k)

Reviewer #2 (Remarks to the Author):

The paper, as revised, addresses all of my major concerns. I didn't think that the other reviewer's concerns were well addressed, but will allow my fellow reviewer to address this. I think the paper is now publishable in this journal.

Reviewer #3 (Remarks to the Author):

The authors seem to have responded very thoroughly to all of my comments, as well as improving the manuscript and the method itself based on all reviewers' comments. While I agree with the comments of reviewer 1 regarding enzyme promiscuity (and repeated some of these criticism myself), I stand by my view that the database filtering applied when generating negative datapoints for the training set should at least mitigate the possibility of erroneously eliminating a true promiscuous activity for an enzyme. It is true that this possibility is not zero, but then, no tool is perfect, and the imperfections of this tool may lead to distinct and useful contradictions between experiments and predictions that can be subsequently scrutinized. So long as the potential flaws in the training set are exceedingly clear and documented, they can be considered as potential sources for prediction discrepancies and be targets for future correction. Hopefully this can be made clear in the discussion.

Regarding novelty, while I agree this paper applies many previously described machine learning techniques, I would argue that the novelty here is in the combination of methods applied, the construction and refinement of the training set, and the integration of negative samples into the training set.

Overall, the paper seems publishable in its current revised form.

Reviewer #4 (Remarks to the Author):

I was asked to comment on the authors' responses to the first report of reviewer 1 and reviewer 1's comments on the revised manuscript.

It seems to me the main issue has shifted to the proposed model having only an incremental improvement in performance, which perhaps is more of an editorial decision on significance.

I am less concerned about the selection of negative examples as I agree with the authors that the validity of their choice is determined by the final results.

The need for more comparison to other approaches may be valid, but only if such approaches exist and provide useful comparisons (I don't know). It seems they did an ablation study, which by itself provides a comparison to different models.

Evaluating on different data splits might be valuable, but it seems like they've done a post-facto analysis by looking at test performance for molecules with different similarities, which should answer the same questions.

I'm not sure what is meant by reviewer 1's statement "It is not standard practice to have a test set and do cross validation at the same time!" If the model was optimized using the test set, that is poor practice. If it was optimized using cross-validation then trained on the full training set and evaluated on a test set, that is standard practice.

Based on reading the reviews, I don't see anything particularly problematic other than perhaps incremental results.

Looking also at the paper, I see no issues with the evaluation - cross-validation was done using the training set and the test set was carefully chosen to be distinct (less than 80% sequence identity) to the training set. This is an appropriate and rigorous evaluation.

Reviewer #1:

Overall concern remains:

- The lack of novelty. ESM-1b was used prior; language models with fingerprints/GNNs were used before; the token is not added information and does not provide significant improvement over not using it. Further, it actually promotes incorrect ideas about how adding redundant information is helpful in machine learning tasks.

Response: We do not argue that the use of the ESM-1b model or the use of molecular fingerprints is novel. The main novelty presented in our manuscript is the development of a general prediction model for specific substrates of enzymes.

The added token is indeed not adding input information for the model; however, we argue that it allows an easier extraction of enzyme information from the trained model. Without the added token, enzyme representations are created by taking the element-wise mean over all amino acid representations, an ad-hoc procedure that likely results in information loss. Importantly, when Transformer Networks are applied to natural language tasks such as text classification, it is standard practice to add such an extra token for the whole input sequence. In this manuscript, we show that the same procedure, applied to protein representations, leads to a small but statistically significant model improvement.

Action: In the revised manuscript, we motivate the addition of an extra token in more detail.

- Evaluation of the model remains limited.
 - No detailed comparison with other methods. By their own admission the comparison provided is not fair. Why not compare against state-of-the-art methods in predicting interactions (similar to ones that are used for drug-target interactions – most of those are downloadable and readily available)

Response: In the section “ESP outperforms two recently published models for predicting the substrate scope of enzymes”, we performed a detailed comparison of our method to two recent methods for predicting substrates of specific enzyme families. It is impossible to execute a completely fair comparison to previous models since ESP is the first model to address the task of developing a general enzyme-substrate prediction model. However, we clearly state this at the beginning of the comparison.

Comparing our model to models developed for completely different tasks - such as drug-target interactions - does not provide any additional value, as these models have been trained on datasets with different structures for different prediction problems.

Action: None

- Not using other datasets to show the generalizability of the model

Response: As stated in the response above and detailed in the original and revised manuscript, we applied ESP to two different datasets (from two recently published models) that were not part of our training set.

Action: None

- The experimental methodology for evaluating the model does not properly evaluate impact of negative data selection nor different data splits, and reporting test data along with cross validation is problematic from a reader's perspective.

Response: The dataset was split into 80% training data and 20% test data. The training set was then split into 5 folds for cross-validation. After performing hyperparameter optimization with a 5-fold CV (only on the training data!), we selected the best model and reported final

model performance on the test set, which had not been used for hyperparameter selection or training. This is common (best) practice in machine learning. In support of our evaluation methodology, we refer to Reviewer #4, who stated “I see no issues with the evaluation - cross-validation was done using the training set, and the test set was carefully chosen to be distinct (less than 80% sequence identity) to the training set. This is an appropriate and rigorous evaluation.”

Moreover, we applied our model to two datasets with experimental evidence for negative data points. The good performance on these independent datasets strongly supports our model’s ability to learn to distinguish between positive and negative data points from training on sampled negative data points.

Furthermore, the results displayed in Figure 4a show the evaluation of the impact of different data splits.

Action: None

Machine learning models are developed to have predictive power. The authors, by their own admission, “conclude that ESP only achieves high accuracies for new enzyme-small molecule pairs if the small molecule was present among the ~1 400 substrates of our training set.”. The model predictive capabilities are therefore limited. From a machine-learning perspective, this should not be the case, provided that the training data is selected more carefully.

Response: Clearly almost every machine learning model has limitations regarding the generalizability to unseen data that is distinct from the training data. We clearly pointed out the limitations of our model and showed when it can be expected to achieve predictions with high accuracies, and in which cases the model has no high predictive power.

Action: None

In response to the specific rebuttals:

1. Selection of the “negative data”. This concern is not addressed, and the justification provided is not scientifically sound.

The authors do not provide sufficient justification WkY their negative sampling methodology is valid.

They state, “This subset comprises only ~1,400 substrates, and most of the metabolites in this biased subset are commonly present in biological cells. “ Looking at their Methods section in the paper, the positive interactions were retrieved from enzyme-substrate pairs listed with catalytic activity in UniProt through RkEA. The ABSENCE OF AN INTERACTION DOES NOT MEAN TkAT IT IS NEGATIVE, NO MATTER KOW POPULAR Tke SUBSTRATE

IS. Further, RkEA, per their website, “Rhea is an expert-curated knowledgebase of chemical and transport reactions of biological interest” – So not clear how extensive their curation efforts are towards promiscuous activities.

So, there has to be a way of sampling the negative data and giving them “less confidence”. Exploring the ratio of the positives and negatives, while balancing the classes would be a route towards addressing this problem.

Response: We do not claim that the sampled substrates can never be indeed positive substrates; instead, we argue in the manuscript why there is a low chance of sampling such false negative data points. Importantly, as pointed out by Reviewer #4, the validity of our sampling process is determined by the model results on independent and new datasets. We emphasize this post-hoc justification of our approach in the manuscript.

Action: None

2a. **The novelty of the ESP model remains a concern based on the author's evaluation.** Thank you for adding the analysis to clarify the impact of this contribution. I am examining your results that you added:

Supplementary Table 4. Results of three gradient boosting models with different enzyme representations. Models were trained with GNN-generated fingerprints as small molecule representations combined with three different enzyme representations: *ESM-1b* vectors, *ESM-1b_{ts}* vectors created without an extra token for the whole enzyme, and *ESM-1b_{ts}* vectors created with an extra token for the whole enzyme. Results are shown for the test set. The hyperparameter optimizations for all models were performed with 5-fold cross-validation on the training set.

	ROC-AUC score	Accuracy	MCC
ESM-1b	0.940	88.8%	0.72
ESM-1b_{ts} (mean representation)	0.956	90.9%	0.77
ESM-1b_{ts} (enzyme token)	0.956	91.5%	0.78

37/42

It looks like your improvement from fine-tuning (a standard practice in machine-learning, not novel) is in the top red box (*ESM-1b_{ts}*) is roughly 2%.

The additional novelty that you are claiming by adding the enzyme token is highlighted in the bottom redbox: that gives you no improvement in ROC-AUC, and less than 1% in Accuracy, and less than 0.01 in MCC, over the fine-tuning method.

The impact of accessing the enzyme token is not significant, per your own results.

From a machine learning perspective, the knowledge in the “enzyme token” should have been already captured by the equivalent representation of the individual amino acids. You are not adding any new information, but summarizing the info for the model ... If the model parameters were further trained and fine-tuned, then it is likely that you would have achieved the additional results without the enzyme token.

Response: Although there is no huge difference between both enzyme representations, we show that the difference in model performance is statistically significant. Moreover, we disagree that the extra enzyme token does not provide any value. As pointed out above, the extra token indeed does not provide any additional input information, but it aids the model in extracting relevant enzyme

information for substrate classification. While the same information is indeed contained in the representations of the individual enzymes, this information is typically summarized in applications of the ESM1b model by averaging over the representations of the individual amino acids. This last summarization step clearly risks losing relevant information, adding an extra token aims to counteract this loss of information. Adding such an extra token is also standard practice when applying Transformer Networks to text classification tasks.

Action: None

3A(a). Concern about training under different data splits was not addressed. This reviewer was not asking for data distribution analysis - they are coming from the same distribution. What was being asked is to consider different data splits to show how the model performs on a more challenging split. There is no need for Supplementary Fig. 1. But please examine the different data splits on the molecules.

Response: As pointed out by Reviewer #4, the analysis of looking at the test performance for different protein sequence identity levels, for different substrate similarities, and for substrate occurrences compared to the test set, does provide the requested insights.

Action: None

3A(b) Thank you for rephrasing the sentence. How it is stated now shows the severe predictive limitations of your models, which is now the main concern regarding this manuscript.

Response: We clearly agree that applying ESP to substrates not included in those ~1400 substrate that were part of our training set is a limitation of our model. However, for those substrates that have been part of the training set, ESP can provide valuable predictions.

Action: None

3(B) It is not standard practice to have a test set and do cross validation at the same time! Following standard machine learning reporting practices would avoid this issue. Please remove the unnecessary explanation and follow standard machine learning reporting practices.

Response: We strongly disagree with this statement. It is standard practice to divide the whole dataset first into a training and a test set as we have done here. Depending on the amount of available training data, the training dataset can then be split again into a training and validation set, or the training set can be split into k folds to perform k-fold cross-validation which we have done in the manuscript. This standard practice is described for example in “Deep Learning with Python” by Francois Chollet (Second Edition, Chapter 4).

Additionally, we would again like to refer to Reviewer #4, who stated “I see no issues with the evaluation - cross-validation was done using the training set and the test set was carefully chosen to be distinct (less than 80% sequence identity) to the training set. This is an appropriate and rigorous evaluation.”

Action: None

3(C) Thank you for addressing the comparison with Mou's. However, with you realizing that it is not a fair comparison, there is currently no fair comparison with any state of the art models on the same dataset. This becomes yet another concern for the

novelty and value of this paper.

Response: In our manuscript, we describe the first general model for predicting enzyme-substrate pairs. Thus, it is impossible to perform a fair comparison to a previous model – no model developed and trained for the same task exists. We do not see how the uniqueness of our approach can be construed as a concern for its novelty.

Action: None

4. Unfortunately, the section of training with increasing data set size remains not interesting as the used dataset is *small* (69,365 interactions), and there is no reason to train with less data. In contrast, the ESM-1b Rives paper trains with **250 MILLION** sequences and that warrants exploring training with less data.

Response: As already pointed out in our response to your previous review, we were not so much interested in exploring how well the model can be trained with less training data. Instead, we trained with different amounts of data to get an impression of how model performance depends on the size of the training data – so that we can estimate how much the model might improve once additional training data becomes available in the future.

Action: None

5. The “k” comes from retrieval metrics in machine learning. A protein in the testset interactions with k molecules. You want to rank the model's ability in ranking these k interactions ahead of all other interactions in the dataset. See wikipedia definitions: [https://en.wikipedia.org/wiki/Evaluation_measures_\(information_retrieval\)#Precision_at_k](https://en.wikipedia.org/wiki/Evaluation_measures_(information_retrieval)#Precision_at_k)

Response: Thank you for providing the link. As stated in that Wikipedia entry, *Precision at k* corresponds to the fraction of true positives among the top k predictions (“e.g., $P@10$ or “Precision at 10” corresponds to the number of relevant results among the top 10 retrieved documents”). The Wikipedia entry suggest using *precision at k* specifically for cases where queries lead to “thousands of relevant documents”. As explained in the Wikipedia entry: “Another shortcoming is that on a query with fewer relevant results than k , even a perfect system will have a score less than 1.” This limitation is highly relevant here: most enzymes in our dataset only have 1 or 2 confirmed positive substrates – so $k \geq 2$ makes no sense for many enzymes. We have to conclude that *Precision at k* is not an appropriate evaluation metric for our model.

Action: None

Reviewer #2:

The paper, as revised, addresses all of my major concerns. I didn't think that the other reviewer's concerns were well addressed, but will allow my fellow reviewer to address this. I think the paper is now publishable in this journal.

Response: We thank the reviewer for the positive assessment of our revised article.

Action: None

Reviewer #3:

The authors seem to have responded very thoroughly to all of my comments, as well as improving the manuscript and the method itself based on all reviewers' comments. While I agree with the comments of reviewer 1 regarding enzyme promiscuity (and repeated some of these criticism myself), I stand by my view that the database filtering applied when generating negative datapoints for the training set should at least mitigate the possibility of erroneously eliminating a true promiscuous activity for an enzyme. It is true that this possibility is not zero, but then, no tool is perfect, and the imperfections of this tool may lead to distinct and useful contradictions between experiments and predictions that can be subsequently scrutinized. So long as the potential flaws in the training set are exceedingly clear and documented, they can be considered as potential sources for prediction discrepancies and be targets for future correction. Hopefully this can be made clear in the discussion.

Regarding novelty, while I agree this paper applies many previously described machine learning techniques, I would argue that the novelty here is in the combination of methods applied, the construction and refinement of the training set, and the integration of negative samples into the training set.

Overall, the paper seems publishable in its current revised form.

Response: We thank the reviewer for this positive assessment. We agree that potential flaws of the sampling process and its room for improvement should be discussed in the discussion.

Action: We further expanded the discussion section according to the Reviewer's suggestion.

Reviewer #4:

I was asked to comment on the authors' responses to the first report of reviewer 1 and reviewer 1's comments on the revised manuscript.

It seems to me the main issue has shifted to the proposed model having only an incremental improvement in performance, which perhaps is more of an editorial decision on significance.

I am less concerned about the selection of negative examples as I agree with the authors that the validity of their choice is determined by the final results.

The need for more comparison to other approaches may be valid, but only if such approaches exist and provide useful comparisons (I don't know). It seems they did an ablation study, which by itself provides a comparison to different models.

Evaluating on different data splits might be valuable, but it seems like they've done a post-facto analysis by looking at test performance for molecules with different similarities, which should answer the same questions.

I'm not sure what is meant by reviewer 1's statement "It is not standard practice to have a test set and do cross validation at the same time!" If the model was optimized using the test set, that is poor practice. If it was optimized using cross-validation then trained on the full training set and evaluated on a test set, that is standard practice.

Based on reading the reviews, I don't see anything particularly problematic other than perhaps incremental results.

Looking also at the paper, I see no issues with the evaluation - cross-validation was done using the training set and the test set was carefully chosen to be distinct (less than 80% sequence identity) to the training set. This is an appropriate and rigorous evaluation.

Response: We appreciate the positive assessment of our revised manuscript.

Action: None